# KTAE: A Model-Free Algorithm to Key-Tokens Advantage Estimation in Mathematical Reasoning

**Wei Sun[1,2], Wen Yang[1,2], Pu Jian[1,2],**
**Qianlong Du[1], Fuwei Cui[1], Shuo Ren[1], Jiajun Zhang[†1,2,3]**
[1]Institute of Automation, Chinese Academy of Sciences
[2]School of Artificial Intelligence, University of Chinese Academy of Sciences
[3]Wuhan AI Research
{sunwei2023,yangwen2023,jianpu2023,shuo.ren,fuwei.cui}@ia.ac.cn
{qianlong.du,jjzhang}@nlpr.ia.ac.cn

**Project Page:** https://github.com/ZNLP/KTAE.git

## Abstract

Recent advances have demonstrated that integrating reinforcement learning with rule-based rewards can significantly enhance the reasoning capabilities of large language models, even without supervised fine-tuning. However, prevalent reinforcement learning algorithms such as GRPO and its variants like DAPO, suffer from a coarse granularity issue when computing the advantage. Specifically, they compute rollout-level advantages that assign identical values to every token within a sequence, failing to capture token-specific contributions and hindering effective learning. To address this limitation, we propose **K**ey-**t**oken **A**dvantage **E**stimation (*KTAE*) - a novel algorithm that estimates fine-grained, token-level advantages without introducing additional models. KTAE leverages the correctness of sampled rollouts and applies statistical analysis to quantify the importance of individual tokens within a sequence to the final outcome. This quantified token-level importance is then combined with the rollout-level advantage to obtain a more fine-grained token-level advantage estimation. Empirical results show that models trained with GRPO+KTAE and DAPO+KTAE outperform baseline methods across five mathematical reasoning benchmarks. Notably, they achieve higher accuracy with shorter responses and even surpass R1-Distill-Qwen-1.5B using the same base model.

## 1 Introduction

Notably, large reasoning language models (LRMs) like OpenAI o1 [1] and DeepSeek R1 [2] have demonstrated the capability to solve complex mathematical reasoning problems that challenge even human experts. This progress marks a significant step toward Artificial General Intelligence (AGI) [3, 4, 5]. These reasoning language models often exhibit behaviors such as self-reflection and self-verification within the reasoning chain, which are critical for enhancing reasoning accuracy. DeepSeek R1 aptly refers to the critical turning points that lead to improved performance as "aha moments". The emergence and cultivation of such moments are greatly facilitated by the application of reinforcement learning (RL) or distilled from more powerful LRMs [6, 7, 2]. For instance, DeepSeek applied RL directly to the base language model, using a simple rule-based reward function to encourage the model to explore and unlock its reasoning potential [8] through self-exploration.

---

[†]Corresponding author.

39th Conference on Neural Information Processing Systems (NeurIPS 2025).

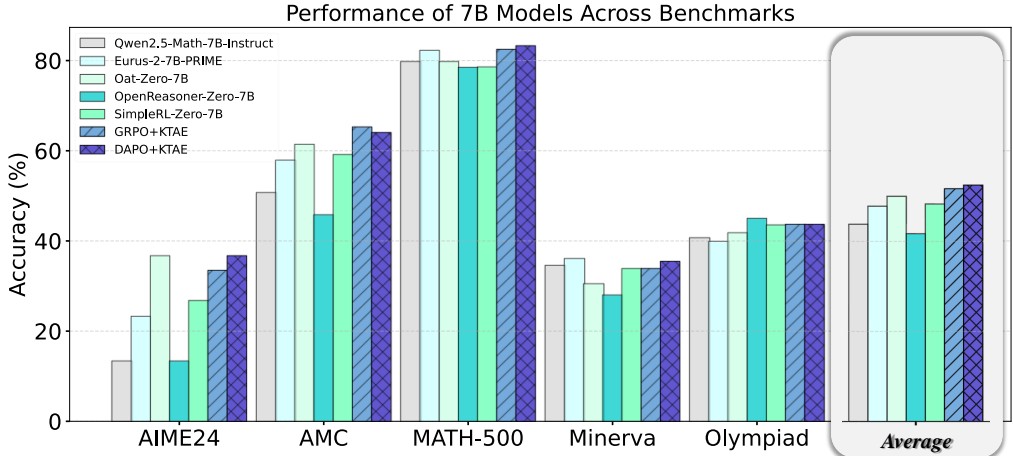

Figure 1: **Model performance comparison**. KTAE is a plug-and-play method that introduces no additional model. It provides token-level advantage estimation for existing RL algorithms such as GRPO and its variants. "GRPO+KTAE" and "DAPO+KTAE" denote GRPO and DAPO combined with KTAE respectively, both RL-tuned on the Qwen2.5-Math-7B model. Detailed results in Table 1.

As a mainstream RL algorithm, Group Relative Policy Optimization (GRPO) [9] differs from the Proximal Policy Optimization (PPO) [10] by eliminating the need for a separate critic model. Instead, it estimates the advantage of each token using the rewards obtained from a set of generated rollouts. However, due to the absence of a critic model, GRPO computes a rollout-level advantage - assigning the same advantage value to every token within a single rollout. This limitation also persists in its improved variant, DAPO [11]. In practice, the importance of each token in a complete Chain-of-Thought (CoT) reasoning sequence varies, and we often observe that incorrect rollouts may only diverge from the correct reasoning path in the final steps. Consequently, applying a uniform advantage value across all tokens in a rollout lacks granularity and may hinder effective learning. Prior efforts have explored using process-level reward models to provide more fine-grained signals [12, 13, 14, 15, 16]. However, as highlighted by DeepSeek [2], training fine-grained reward models is costly, difficult to scale, has limited capacity to provide accurate signals, and is prone to reward hacking [17].

To address these challenges, we propose the **K**ey-**t**oken **A**dvantage **E**stimation (**KTAE**) algorithm. KTAE introduces no additional models, and instead leverages the correctness of sampled rollouts and the occurrence of each token within them to construct a contingency table. Then, using statistical methods such as Fisher's exact test and Information Gain (IG), it quantifies the strength of association between each token and correct rollout. Subsequently, by combining the token's frequency and the reward assigned to its corresponding rollout, KTAE further quantifies the direction of this association's contribution. Finally, these measures are combined (e.g., through multiplication) to yield a 'key-token value' for each token. As shown in Figure 2, when a correct rollout is incorrectly classified as incorrect by the rule, KTAE can still highlight the positively contributing tokens through computing the key-token values. In contrast, GRPO assigns the same negative advantages to all tokens in such a case. Moreover, KTAE can effectively distinguish between tokens irrelevant to problem solving, such as 'First' and 'denote', and those highly relevant to problem solving, such as 'complement' and 'ratio'. Furthermore, KTAE is compatible with GRPO and DAPO. The resulting key-token values are then added to the rollout-level advantage computed by GRPO to obtain a more fine-grained token-level advantage estimate. As illustrated in Figure 1, integrating KTAE with either GRPO or DAPO yields improved performances on average across five major mathematical reasoning benchmarks. Moreover, KTAE not only improves test accuracy but also very effectively reduces response length without any length penalty reward, resulting in extremely high reasoning efficiency.

In summary, the KTAE algorithm offers several advantages:

1. KTAE provides more fine-grained advantage information without introducing extra models, resulting in lower training costs.

2. KTAE quantifies the importance differences between tokens using statistical analysis methods, offering strong interpretability.

3. KTAE's key-token value is computed based on the correctness of the final answer and retains the original rollout-level advantage, making it less susceptible to reward hacking.

4. KTAE helps the model to focus on key tokens and reduce the learning of irrelevant tokens, which can effectively reduce the response length.

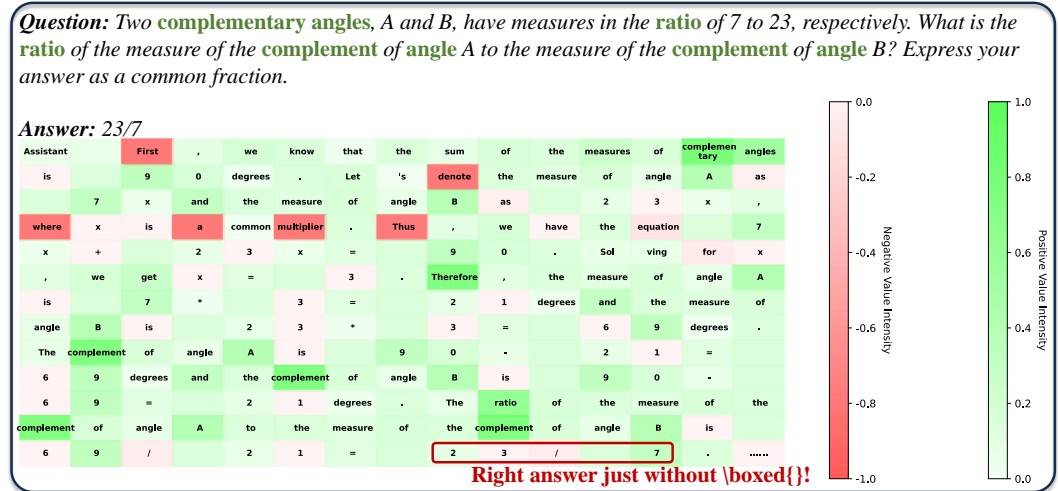

Figure 2: Visualization of key-token-values computed by KTAE for a correct rollout whose final result was unparsable and thus received a final reward of 0. Red shading indicates negative token associations with producing a correct rollout, with darker red representing stronger negative influence; Green shading indicates positive associations.

## 2 Preliminary

### 2.1 Reinforcement Learning in LLM

When applying Reinforcement Learning (RL) to language models, text generation is modeled as a token-level Markov Decision Process (MDP). At each timestep $t$, the state $s_t$ consists of the input prompt $q$ and the previously generated tokens $[o_1, \ldots, o_{t-1}]$, i.e., $s_t = [q; o_1, \ldots, o_{t-1}]$. The policy $\pi$ generates the next token $o_t$ as action $a_t$, and generation ends upon producing an end-of-sequence token or reaching a maximum length $T$. The full sequence $o = [o_1, \ldots, o_T]$ is then evaluated by a reward function $R(q, o) = \sum_{t=1}^{|o|} r(s_t, o_t)$. The RL objective aims to maximize an entropy-regularized expected reward [18]:

$$\mathcal{J}(\pi_\theta) = \mathbb{E}_{(q,o)\sim\pi}[R(q, o)] - \beta\mathbb{E}_{q\sim D, s_t\sim\pi_\theta}[D_{KL}(\pi_\theta(\cdot|s_t)||\pi_{ref}(\cdot|s_t))] \tag{1}$$

Here, $\pi_{ref}$ is a reference policy, $D_{KL}$ denotes the KL divergence, and $\beta$ controls the penalty strength. This KL term, central to RLHF, discourages large shifts from the reference distribution to preserve fluency and diversity. In recent mathematical reasoning tasks [11, 19], $\beta$ is typically set to 0.

### 2.2 GRPO

Group Relative Policy Optimization (GRPO) [9] is simplified based on PPO and eliminates the need for a value model. Given an input $q$, GRPO samples $G$ rollouts $\{o_1, \ldots, o_G\}$ from the old policy and computes their cumulative rewards $R = \{R_1, \ldots, R_G\}$. These rewards are then used to estimate advantages $\hat{A}_{i,t}$, e.g., by comparing each $R_i$ to a baseline derived from $R$. The optimization objective for GRPO is defined as follows:

$$\mathcal{J}_{\text{GRPO}}(\theta) = \mathbb{E}_{q\sim\mathcal{D}, \{o_i\}_{i=1}^G\sim\pi_{\theta_{\text{old}}}(\cdot|q)}$$

$$\left[\frac{1}{G}\sum_{i=1}^{G}\frac{1}{|o_i|}\sum_{t=1}^{|o_i|}\left(\min\left(r_{i,t}(\theta)\hat{A}_{i,t}, \text{clip}\left(r_{i,t}(\theta), 1-\varepsilon, 1+\varepsilon\right)\hat{A}_{i,t}\right) - \beta D_{\text{KL}}(\pi_\theta||\pi_{\text{ref}})\right)\right] \tag{2}$$

where $r_{i,t}(\theta) = \frac{\pi_\theta(o_{i,t}|q,o_{i,<t})}{\pi_{\theta_{old}}(o_{i,t}|q,o_{i,<t})}$, and $\hat{A}_{i,t}$ is the advantage estimate derived from the group rewards $R$, defined as $\hat{A}_{i,t} = \frac{R_i - \text{mean}(\mathbf{R})}{\text{std}(\mathbf{R})}$. The clipping term with clip ratio $\varepsilon$ [20] aims to constrain the new policy within the trust region of the old policy, enhancing training stability. By eliminating the dependency on the value model $V_\phi$, GRPO aims to substantially reduce training costs while striving to maintain optimization effectiveness comparable to traditional PPO.

## 2.3 DAPO

Dynamic Sampling Policy Optimization (DAPO) [11] is an enhancement algorithm of GRPO, specifically tailored for tasks involving mathematical reasoning. To mitigate the phenomenon of entropy collapse, DAPO introduces the "Clip-Higher" method, which raises the upper bound of the clipping function. It incorporates "Dynamic Sampling" to prevent scenarios where all G sampled rollouts exhibit identical preference outcomes (e.g., all positive or all negative). A "Token-Level Policy Gradient Loss" is employed to stabilize the training process. Additionally, DAPO introduces "overlong reward shaping" to penalize excessively long responses, thereby preventing the model from falling into catastrophic repetition loops.

$$
\begin{aligned}
\mathcal{J}_{\text{DAPO}}(\theta) = \quad & \mathbb{E}_{q \sim \mathcal{D}, \{o_i\}_{i=1}^G \sim \pi_{\theta_{old}}(\cdot|q)} \\
& \left[ \frac{1}{\sum_{i=1}^G |o_i|} \sum_{i=1}^G \sum_{t=1}^{|o_i|} \min\left( r_{i,t}(\theta)\hat{A}_{i,t}, \text{clip}\left(r_{i,t}(\theta), 1 - \varepsilon_{\text{low}}, 1 + \varepsilon_{\text{high}}\right)\hat{A}_{i,t} \right) \right] \quad (3) \\
\text{s.t.} \quad & 0 < \left| \{o_i \mid \texttt{is\_equivalent}(a, o_i)\} \right| < G,
\end{aligned}
$$

where $r_{i,t}(\theta)$ and $\hat{A}_{i,t}$ are the same as GRPO. $\varepsilon_{\text{low}}$ and $\varepsilon_{\text{high}}$ represent the upper and lower bounds of $r_{i,t}$ after decoupling.

# 3 KTAE: A Model-Free Algorithm to Key-Tokens Advantage Estimation

GRPO's advantage estimation has a relatively coarse granularity. It assigns the same advantage value to every token within the same rollout. However, in tasks requiring complex reasoning steps, such as mathematical reasoning, the importance of different tokens within a rollout can vary significantly. To address this, we propose the KTAE (Key-tokens Advantage Estimation) algorithm. Without additional models, KTAE quantifies the importance of different tokens by analyzing the statistical associations within the set of sampled rollouts (correct vs. incorrect). It then integrates this quantified token importance with rollout-level advantage estimates (computed by GRPO) to produce fine-grained, token-level advantage estimations. In this section, we will introduce its calculation process in detail.

## 3.1 Building Token-Level Contingency Tables

For a given problem, we sample a set of $G$ rollouts $\{o_1, \ldots, o_G\}$, each with a corresponding rule-based reward $\{R_1, \ldots, R_G\}$ indicating its correctness, following the same approach as GRPO. We divide these rollouts into a correct set $x_{\text{T}}$ and an incorrect set $x_{\text{F}}$. For each token $o_{ij}$ in the sampled rollouts $o_i$, we examine its occurrence across all $G$ rollouts and construct a $2 \times 2$ contingency table summarizing the counts of correct and incorrect rollouts that contain or do not contain $o_{ij}$. An example contingency table is shown in Figure 3.

Here, $G$ is the total number of rollouts. For token $o_{ij}$, we use the statistics $a_{o_{ij}}, b_{o_{ij}}, c_{o_{ij}}, d_{o_{ij}}$ based on its occurrence across the rollouts sets. $a_{o_{ij}}$ is the count of rollouts in $x_{\text{T}}$ where $o_{ij}$ appears at least once, $a_{o_{ij}} = \sum \mathbb{I}(o_{ij} \in x_{\text{T}})$; $c_{o_{ij}}$ is the count of rollouts in $x_{\text{T}}$ where $o_{ij}$ does not appear $c_{o_{ij}} = sum(x_{\text{T}}) - a_{o_{ij}}$; $b_{o_{ij}}$ and $d_{o_{ij}}$ are calculated in the same way. The total count: $a_{o_{ij}} + b_{o_{ij}} + c_{o_{ij}} + d_{o_{ij}} = G$. We then use the statistics from this contingency table to compute the association between the occurrence of $o_{ij}$ and the policy sampling a correct rollout.

## 3.2 Quantifying Association Strength via Hypothesis Testing

We begin by quantifying the association using hypothesis testing. We set the null hypothesis ($H_0$) as: the occurrence of $o_{ij}$ and the correctness of its rollout have no association. We use Fisher's exact

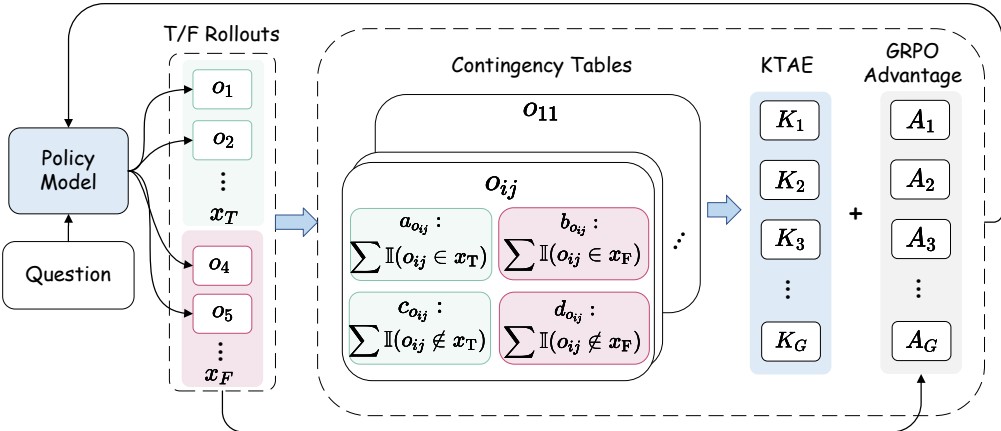

Figure 3: **The outline of KTAE algorithm.** It constructs a contingency table based on the correctness of the sampled rollouts, and then calculates the token-level advantage and adds it to the GRPO's rollout-level advantage.

test [21] to compute the $p$-value, which is the probability of observing the current contingency table or a more extreme one, assuming $H_0$ is true. The formula for Fisher's exact test $p$-value is:

$$Fisher(o_{ij}) = \frac{\binom{a_{o_{ij}}+b_{o_{ij}}}{a_{o_{ij}}}\binom{c_{o_{ij}}+d_{o_{ij}}}{c_{o_{ij}}}}{\binom{N}{a_{o_{ij}}+c_{o_{ij}}}} = \frac{(a_{o_{ij}}+b_{o_{ij}})!(c_{o_{ij}}+d_{o_{ij}})!(a_{o_{ij}}+c_{o_{ij}})!(b_{o_{ij}}+d_{o_{ij}})!}{a_{o_{ij}}!b_{o_{ij}}!c_{o_{ij}}!d_{o_{ij}}!N!} \quad (4)$$

In practice, this calculation is performed in log-space to handle large factorials (see appendix D for details). A smaller $p$-value indicates stronger evidence against the null hypothesis, meaning a stronger association between the occurrence of token $o_{ij}$ and rollout correctness. Since effective $p$-values are often concentrated in a small range, we employ a transformation function to quantify the association strength and amplify the impact of small $p$-values. We define the association score of Fisher's test as:

$$\mathcal{F}(o_{ij}) = \begin{cases} e^{-2 \cdot Fisher(o_{ij})} & \text{if } Fisher(o_{ij}) \neq 1 \\ 0 & \text{if } Fisher(o_{ij}) = 1 \end{cases} \quad (5)$$

When $p = 1$ (complete no association), the score is 0; when $p$ approaches 0 (strong association), the score approaches 1. Fisher's exact test is chosen over chi-squared or G-tests because the latter provide less accurate $p$-values for small sample sizes $N$, while Fisher's test offers an exact probability calculation even with small $G$ (e.g., $G = 8$ or 16).

### 3.3 Quantifying Association Strength via Information Gain

To complement the statistical test with an information-theoretic perspective, we compute the Information Gain (IG) between the occurrence of the token $o_{ij}$ and rollout correctness. Let $Y$ be a random variable representing rollout correctness, and $X_{o_{ij}}$ be a variable indicating whether the token $o_{ij}$ appears in a rollout. The entropy of rollout correctness $H(Y)$ is:

$$H(Y) = -\frac{a_{o_{ij}} + c_{o_{ij}}}{N} \log_2\left(\frac{a_{o_{ij}} + c_{o_{ij}}}{N}\right) - \frac{b_{o_{ij}} + d_{o_{ij}}}{N} \log_2\left(\frac{b_{o_{ij}} + d_{o_{ij}}}{N}\right) \quad (6)$$

The conditional entropy of rollout correctness given whether token $o_{ij}$ appears, $H(Y|X_{o_{ij}})$, is:

$$\begin{aligned} H(Y|X_{o_{ij}}) = & \left(\frac{a_{o_{ij}} + b_{o_{ij}}}{N}\right)\left[-\frac{a_{o_{ij}}}{a_{o_{ij}} + b_{o_{ij}}}\log_2\left(\frac{a_{o_{ij}}}{a_{o_{ij}} + b_{o_{ij}}}\right) - \frac{b_{o_{ij}}}{a_{o_{ij}} + b_{o_{ij}}}\log_2\left(\frac{b_{o_{ij}}}{a_{o_{ij}} + b_{o_{ij}}}\right)\right] \\ & + \left(\frac{c_{o_{ij}} + d_{o_{ij}}}{N}\right)\left[-\frac{c_{o_{ij}}}{c_{o_{ij}} + d_{o_{ij}}}\log_2\left(\frac{c_{o_{ij}}}{c_{o_{ij}} + d_{o_{ij}}}\right) - \frac{d_{o_{ij}}}{c_{o_{ij}} + d_{o_{ij}}}\log_2\left(\frac{d_{o_{ij}}}{c_{o_{ij}} + d_{o_{ij}}}\right)\right] \end{aligned} \quad (7)$$

The Information Gain (IG) is defined as $IG(o_{ij}) = H(Y) - H(Y|X_{o_{ij}})$.

A higher $IG$ value indicates that knowing whether the token $o_{ij}$ appears reduces the uncertainty about rollout correctness more significantly, suggesting a stronger association with correctness. Otherwise, means the association is weaker. Through Fisher's exact test and Information Gain, we have quantified the strength of association between the occurrence of token $o_{ij}$ and rollout correctness (e.g., via a linear combination $h_1 \cdot \mathcal{F}(o_{ij}) + h_2 \cdot IG(o_{ij})$).

### 3.4  Quantifying Association Direction and Final Importance Score

However, both $\mathcal{F}(o_{ij})$ and $IG(o_{ij})$ can only quantify the strength of association between the occurrence of $o_{ij}$ and the correct rollout, they cannot quantify the direction of this association (i.e., positive or negative association). For the detailed proof, see Appendix E. To determine the direction of the association and further quantify token importance, we adapt the Term Frequency calculation idea from BM25 [22] to compute standardized frequency scores for the token $o_{ij}$ within the set of correct and incorrect rollouts.

Specifically, we concatenate all correct rollouts into a single long sequence and all incorrect rollouts into another. We compute the term frequency (tf) of the token $o_{ij}$ in these two concatenated sequences, denoted as $\text{tf}_T(o_{ij})$ and $\text{tf}_F(o_{ij})$. Based on tfs we compute standardized frequency scores:

$$TF_{T/F}(o_{ij}) = \frac{(k_1 + 1) \cdot \text{tf}_{T/F}(o_{ij})}{k_1(1 - b + b \times \frac{len_{T/F}}{len_{\text{avg}}}) + \text{tf}_{T/F}(o_{ij})} \tag{8}$$

Here, T/F refers to the correct or incorrect rollouts. $len_T$ and $len_F$ are the average lengths of the concatenated correct and incorrect rollouts, respectively, and $len_{\text{avg}}$ is the average rollout length across all $G$ rollouts. $k_1$ and $b$ are adjustable parameters controlling the influence of term frequency and length normalization (can be set empirically or tuned). Treating all correct/incorrect rollouts as single sequences reduces the impact of individual rollouts with extreme lengths.

The token directional score $D(o_{ij})$ combines a measure of effect size based on proportion differences and a measure based on standardized frequency score differences. We use Cohen's h effect size ($\arcsin \sqrt{x} - \arcsin \sqrt{y}$) metric to measure the difference in the proportion of correct rollouts ($\frac{a_{o_{ij}}}{a_{o_{ij}} + c_{o_{ij}}}$) versus incorrect rollouts ($\frac{b_{o_{ij}}}{b_{o_{ij}} + d_{o_{ij}}}$) where $o_{ij}$ appears. Simultaneously, we consider the ratio difference of the standardized frequency scores. The final formula is:

$$D(o_{ij}) = \left( \arcsin \sqrt{\frac{a_{o_{ij}}}{a_{o_{ij}} + c_{o_{ij}}}} - \arcsin \sqrt{\frac{b_{o_{ij}}}{b_{o_{ij}} + d_{o_{ij}}}} \right) + h_3 \left( \frac{TF_T(o_{ij})}{TF_F(o_{ij})} - \frac{TF_F(o_{ij})}{TF_T(o_{ij})} \right) \tag{9}$$

This combination aims to capture different aspects of importance: when the token $o_{ij}$'s frequency is similar in correct and incorrect rollouts(High frequency generic tokens), its importance might be better reflected by the probability difference in where it appears, hence the dominance of the arcsin square root proportion difference term ( Cohen's h effect size); when the token $o_{ij}$'s frequency differs significantly (especially for low-frequency but critical tokens), the frequency ratio better reflects its discriminative power, increasing the importance of the frequency score ratio term.

Theoretically, both the Fisher score $\mathcal{F}(o_{ij})$ and the Information Gain $IG(o_{ij})$ are strictly greater than zero, while the directional score $D(o_{ij})$ spans the full real range $(-\infty, +\infty)$. To derive the final token-level relevance score, we multiply the magnitude of correlation (e.g., $\mathcal{F}(o_{ij})$ or $IG(o_{ij})$) by the directionality score $D(o_{ij})$, which reflects whether the token is positively or negatively associated with correct rollouts. Finally get key-token-value of $o_{ij}$ is $(h_1 \cdot \mathcal{F}(o_{ij}) + h_2 \cdot IG(o_{ij})) \cdot D_{o_{ij}}$. Positive key-token-values represent positive association direction.

To stabilize training and constrain the output range, we apply a sigmoid normalization to the resulting key-token-values. These normalized values are then added to the rollout-level advantage computed by GRPO, thereby producing the final token-level advantage:

$$\hat{A}_{o_{ij}}^{KTAE} = \hat{A}_{o_i}^{GRPO} + \sigma((h_1 \cdot \mathcal{F}(o_{ij}) + h_2 \cdot IG(o_{ij})) \cdot D_{o_{ij}}) - 0.5 \tag{10}$$

KTAE is an algorithm for estimating the advantage of tokens, which computes the key-token-value through the rollouts obtained from sampling. The complete implementation process is shown in Algorithm 1. It is orthogonal to the improvement strategy of DAPO, and can be combined with DAPO in addition. An schematic diagram is shown in Appendix F.

---

**Algorithm 1** Key-token Advantage Estimation(KTAE)

---

**Input:** Set of $G$ rollouts $\{o_1, \ldots, o_G\}$ sampled from policy model, rule-based reward $\{R_1, \ldots, R_G\}$, weighting parameter $h_1, h_2, h_3$

1: Calculate the rollout-level advantage of GRPO $\hat{A}^{GRPO}$
2: Summarize all tokens in rollouts into a set $O$
3: Divide the G rollouts into $x_T$ and $x_F$ sets according to the reward $R$ obtained by each rollout
4: **for** $o$ in $O$ **do**
5:    $a = \sum \mathbb{I}(o_{ij} \in x_T)$, $b = \sum \mathbb{I}(o_{ij} \in x_F)$, $c = Len(x_T) - a$, $d = Len(x_F) - b$
6:    Calculate $\mathcal{F}(o)$ according to Eq. 4, and calculate $IG(o)$ according to Eq. 6 and Eq. 7
7:    Weighted add $\mathcal{F}(o)$ and $IG(o)$ to get quantized association strength $h_1 \cdot \mathcal{F}(o) + h_2 \cdot IG(o)$
8:    Calculate the frequency of $o$ in the correct and incorrect rollouts according to Eq. 8
9:    Calculate the quantized association direction according to Eq. 9
10:    Multiply the association direction and association strength to get the key-token-value of each token, and then add it to $\hat{A}_o^{GRPO}$ to get $\hat{A}_o^{KTAE}$
11: **end for**
**Output:** $\hat{A}^{KTAE}$

---

## 4    Experiment

**Experiment Setting.**    Our validation and ablation experiments were conducted on the Qwen2.5-Math-1.5B [23] base model and the comparison experiment with baseline methods is based on Qwen2.5-Math-7B base model, using math12k [12] and its subset math-level3-5 respectively. See Appendix G.1 for specific details of dataset and benchmark, Appendix G.3 for implementation details and hyperparameters, and Appendix J for prompt details.

**Method Validation Result.**    Experiments revealed several key performance trends (Fig. 4). KTAE consistently enhances MATH500 test accuracy when integrated with GRPO and DAPO, respectively. Regarding mean response length, the addition of KTAE significantly reduced the response length for both algorithms compared to their original versions. We believe that achieving improved model performance while simultaneously reducing generation cost is more meaningful. In terms of generation entropy, GRPO+KTAE showed accelerated entropy decrease early on but stabilized at a higher level later, beneficial for mitigating entropy collapse [11]. For DAPO+KTAE, its entropy value was considerably higher than all other configurations and exhibited a continuous upward trend. While such high entropy contributes to increased sampling diversity and avoids entropy collapse, it may also introduce a potential risk of reduced training stability.

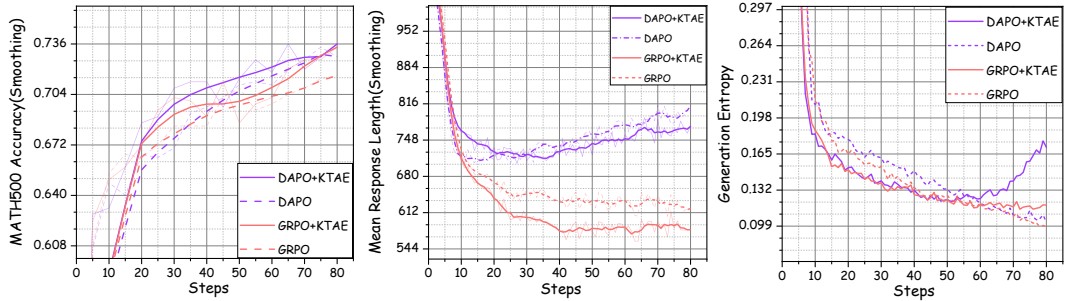

Figure 4: The metric curves of test accuracy, mean response length, and generation entropy of combining DAPO and GRPO with KTAE.

**Comparison with Baselines.**    In Table 1, the DAPO+KTAE-7B model achieved the highest average score across the 5 benchmarks, significantly outperforming others on MATH500. Both GRPO+KTAE and DAPO+KTAE achieved better performance than all baseline methods on AMC (See Appendix G.2 for more details about baselines). GRPO+KTAE showed performance improvements compared to the original GRPO on four out of five benchmarks, with only a slight decrease on AIME24 (Considering AIME24 has just 30 problems, this equates to only one fewer correct answer). Likewise,

compared to the original DAPO, DAPO+KTAE's performance improved or remained unchanged on four out of five benchmarks, experiencing a slight decrease only on OlympiadBench. This demonstrates the effectiveness of the KTAE algorithm. This performance was consistent with the 1.5B model, where our model even surpassed R1-Distill-Qwen-1.5B with the same base model.

Table 1: The zero-shot greedy pass@1 performance of the 1.5B and 7B models across five mathematical reasoning benchmarks. All the results above are of our reproduction. * refers to OlympiadBench; † denotes the results from [24]. ‡ denotes the results from [25], @8k refers the max response length.

| 1.5B Models | AIME24 | MATH-500 | AMC | Minerva | Olympiad* | Avg |
|---|---|---|---|---|---|---|
| Qwen2.5-Math-1.5B-Instruct [‡] | 10.0 | 74.2 | 48.2 | 26.5 | 40.2 | 39.8 |
| Qwen2.5-Math-1.5B [‡] | 16.7 | 61.8 | 43.4 | 15.1 | 28.4 | 33.1 |
| R1-Distill-Qwen-1.5B@8k [‡] | 20.0 | 77.4 | 49.4 | 25.0 | 35.8 | 41.5 |
| Oat-Zero-1.5B [25] | 20.0 | 74.4 | 50.6 | 23.9 | 37.0 | 41.2 |
| GRPO-1.5B | 16.7 | 76.0 | 51.8 | 22.1 | 36.3 | 40.6 |
| **GRPO+KTAE-1.5B** | 26.7 | 75.4 | 41.0 | 27.2 | 38.2 | 41.7 |
| DAPO-1.5B | 16.7 | 77.6 | 47.0 | 25.7 | 39.0 | 41.2 |
| **DAPO+KTAE-1.5B** | 20.0 | 77.6 | 50.6 | 29.0 | 40.0 | 43.4 |
| **7B Models** | AIME24 | MATH-500 | AMC | Minerva | Olympiad* | Avg |
| Qwen2.5-Math-Instruct [26] [†] | 13.3 | 79.8 | 50.6 | 34.6 | 40.7 | 43.8 |
| Qwen2.5-Math [†] | 13.3 | 57.6 | 45.0 | 14.7 | 23.7 | 30.9 |
| Eurus-2-7B-PRIME [27] | 23.3 | 82.2 | 57.8 | 36.0 | 39.9 | 47.8 |
| Oat-Zero-7B [25] | 36.7 | 79.8 | 61.4 | 30.5 | 41.8 | 50.0 |
| OpenReasoner-Zero-7B [19] | 13.3 | 78.4 | 45.8 | 27.9 | 45.0 | 41.7 |
| SimpleRL-Zero-7B [28] | 26.7 | 78.6 | 59.0 | 33.8 | 43.4 | 48.3 |
| GRPO-7B | 36.7 | 81.0 | 57.8 | 32.7 | 43.2 | 50.3 |
| **GRPO+KTAE-7B** | 33.3 | 82.4 | 65.1 | 33.8 | 43.7 | 51.7 |
| DAPO-7B | 36.7 | 81.8 | 60.2 | 34.5 | 45.3 | 51.7 |
| **DAPO+KTAE-7B** | 36.7 | 83.2 | 63.9 | 35.3 | 43.7 | 52.5 |

Table 2 demonstrates that our model can also significantly reduces the length of the response without any length penalty reward. This effect is particularly pronounced for the 7B parameter model, where the GRPO+KTAE model exhibits a considerably shorter generation lengths compared to the baseline methods. This indicates that the KTAE algorithm enables the model to concentrate more effectively on key tokens that are crucial for problem resolution, thereby curtailing the generation of redundant or non-essential tokens. That is to say, KTAE achieved the highest average score across the 5 benchmarks while using the least token budget, demonstrating the highest reasoning efficiency.

As Table 3, KTAE is a model-free algorithm whose computational cost is largely independent of model size, depending primarily on the number of generated tokens. For larger models such as the 7B variant, each training step is inherently slower, making the relative efficiency loss introduced by KTAE less noticeable. In contrast, for smaller models like the 1.5B variant, the shorter training steps make KTAE's overhead more apparent. Nevertheless, the computational cost of KTAE remains acceptable compared to methods that rely on model-based computation.

In practice, KTAE's runtime mainly depends on implementation efficiency. A CPU-only serial implementation would require several hours, as the current version employs only limited tensor parallelism across N rollouts per sampling, resulting in low data parallelism and GPU utilization below 1%. Further optimization—such as concatenating computations into larger tensors to better exploit GPU capabilities—could substantially improve efficiency. We plan to pursue these engineering optimizations in future work.

**Ablation Analysis.** Figure 5 shows the impact of each KTAE component. Removing any component consistently reduced test accuracy. Excluding $IG$ had the largest negative effect on accuracy and

Table 2: The response length of the 1.5B and 7B models across five mathematical reasoning benchmarks. All the results above are of our reproduction. * refers to OlympiadBench.

| 1.5B Models | AIME24 | MATH-500 | AMC | Minerva | Olympiad* | Avg |
|---|---|---|---|---|---|---|
| Oat-Zero-1.5B | 1198 | 878 | 652 | 692 | 938 | 871.6 |
| GRPO-1.5B | 1299 | 635 | 908 | 731 | 958 | 906.2 |
| **GRPO+KTAE-1.5B** | 1187 | 884 | 617 | 663 | 890 | 848.2 |
| DAPO-1.5B | 1218 | 617 | 950 | 712 | 937 | 886.8 |
| **DAPO+KTAE-1.5B** | 1110 | 983 | 582 | 666 | 861 | 840.4 |
| **7B Models** | AIME24 | MATH-500 | AMC | Minerva | Olympiad* | Avg |
| Eurus-2-7B-PRIME | 1498 | 685 | 1099 | 777 | 1077 | 1027.2 |
| Oat-Zero-7B | 977 | 658 | 903 | 677 | 892 | 821.4 |
| OpenReasoner-Zero-7B | 2300 | 1193 | 1901 | 1269 | 1871 | 1706.8 |
| SimpleRL-Zero-7B | 1074 | 634 | 832 | 584 | 881 | 801 |
| GRPO-7B | 989 | 606 | 806 | 641 | 813 | 771.0 |
| **GRPO+KTAE-7B** | 941 | 563 | 741 | 577 | 771 | 718.6 |
| DAPO-7B | 1155 | 676 | 969 | 700 | 986 | 897.2 |
| **DAPO+KTAE-7B** | 1013 | 604 | 864 | 607 | 798 | 777.2 |

Table 3: Comparison of the training time of one step of the KTAE algorithm and the baseline algorithm under the same hardware conditions (8 NVIDIA H100 80G)

| Algorithms | Qwen2.5-7B-MATH | Qwen2.5-1.5B-MATH |
|---|---|---|
| GRPO | 559.36s | 277.69s |
| GRPO+KTAE | 641.98s | 371.79s |
| DAPO | 1006.30s | 363.36s |
| DAPO+KTAE | 1159.00s | 642.74s |

produced the shortest sequences. In contrast, removing $\mathcal{F}$ or $tf$ decreased accuracy while increasing sequence length, though lengths remained shorter than those of the GRPO baseline. For entropy, removing $tf$ initially caused a notable increase. Importantly, while GRPO suffered from entropy collapse, KTAE avoided this. Overall, $IG$ is key for accuracy and brevity, $tf$ supports diversity and stability, and $\mathcal{F}$ contributes to overall performance. All components are essential in accuracy.

**Visualization Example.** Beyond the example illustrated in Figure 2, we also observed several zero-reward rollouts. These rollouts are characterized by including the correct answer in their early stages, but subsequently generating a large number of redundant tokens, ultimately leading to the correct answer being obscured by the subsequent sequence. Appendix H provides a such example, where a clear boundary can be distinctly observed, effectively separating the correct answer from the redundant tokens. This further validates KTAE's accuracy in identifying key tokens. As shown in Appendix K, we also observed the 'aha moment' phenomenon [2] during the KTAE training process.

## 5    Related Work

**Large Reasoning Models.** Breakthroughs [1, 2, 3, 4, 29, 30] in Large Language Models (LLMs) enable a new era of test-time scaling [31, 32] and human-like, stepwise reasoning. DeepSeek R1 [2] used pure RL to induce long Chain-of-Thought (CoT) and self-reflection. Following R1, subsequent work [11, 19, 28, 27, 25, 33] explored RL training variants, mainly on smaller models. While R1's paradigm was replicated [11, 19, 28, 27, 25], exploring more fine-grained reward in GRPO is challenging. This work proposes token-level advantage estimation for GRPO and its variants.

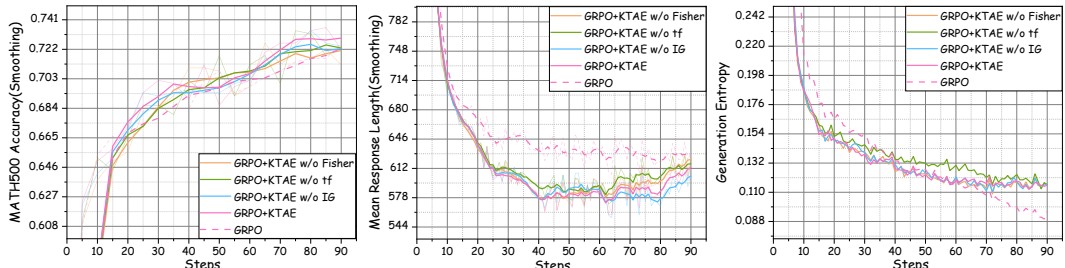

Figure 5: Training results after removing different components from KTAE.

**Reinforcement Learning.** RL is key for sequential decision-making, using policy gradient methods. Early methods (e.g., REINFORCE [34], DPO [35]) had high variance. TRPO [20] and PPO [10] improved stability with constrained/clipped updates, though PPO is costly. GRPO [9] removed the Critic using group statistics. GRPO variants, like DAPO [11] and Dr.GRPO [25], built on this. However, GRPO and variants use uniform rollout advantage, ignoring token importance in reasoning. We propose Key-token Advantage Estimation, linking tokens to correctness statistically, for finer granularity. More related work in Appendix I

## 6 Conclusion

This paper introduces KTAE, an algorithm uses statistical analysis to quantify each token's association to correct rollouts. By combining this with GRPO's rollout-level advantage, KTAE computes token-level advantages, thereby providing more fine-grained optimization signals and significantly improving training effectiveness. It requires no new models, adds minimal computational overhead, and avoids reward hacking. KTAE can effectively identify the importance of different tokens in the rollout, making the model pay more attention to key tokens in the training process, showing excellent test performance utilizing the minimum token budget. Theoretically, the core idea of the KTAE can be applied to many other reasoning domains. Therefore, KTAE still holds significant potential.

## 7 Acknowledgements

We thank our colleagues Jianghao Chen, Chong Li, Tengxiao Xi, Tianyu Peng, Xingquan Zhang, Boyu Guan, Jiawei Guo for their insightful and constructive feedback. We thank Qian Li and Zhenggang Piao for their special assistance. In addition, we thank all reviewers for their valuable comments and recognition of our work. This work is supported by National Key R&D Program of China 2022ZD0160602 and the Strategic Priority Research Program of Chinese Academy of Sciences under Grant XDA04080400.

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

# Appendix

## A  Limitations

Building upon GRPO, the KTAE algorithm introduces a more fine-grained token-level advantage, which guides the model to focus more on key tokens, thereby demonstrating significant performance improvement without introducing additional models. However, our experimental validation is primarily focused on models with 1.5B and 7B parameters, and the performance of KTAE on larger-scale models has not yet been fully verified. Furthermore, while theoretically KTAE is applicable to any task beyond based on rule-based rewards, this paper only provides in-depth analysis and experimental validation on mathematical reasoning tasks, and its potential for application in broader domains requires further exploration.

## B  Broader Impacts

Mathematics represents a pinnacle of human wisdom and serves as the foundation of many scientific disciplines. Our approach aims to empower large language models to tackle complex mathematical reasoning problems, bringing their capabilities closer to human expert-level intelligence. By doing so, we seek to advance the development of large language models across scientific fields and support human efforts in driving scientific progress. A common limitation among current large reasoning language models is that while their reasoning capabilities are enhanced, they tend to sacrifice some general abilities like summarization, abstracting, and translation. Consequently, these models may become specialized models primarily focused on the reasoning domain.

## C  Future Work

The experiments presented in this paper primarily focus on tasks based on binary discrete rewards, such as in the fields of mathematics and code generation, where the reward is simply classified as either correct or incorrect. Future work will explore how to extend the core idea of KTAE to scenarios with multiclass discrete rewards, and even to those with continuous reward values.

Furthermore, although KTAE quantifies the importance of each token to provide fine-grained optimization signals, the amount of information carried by a single token is often insufficient within a complete reasoning path. This implies that, even for human experts, it is difficult to determine whether the presence of a single token has a decisive impact on the final outcome (correct or incorrect), or whether its absence would necessarily lead to failure. Future work will aim to address or optimize this problem, for instance.

## D  Calculating factorials using logarithmic space

The Gamma function is defined by the following integral:

$$\Gamma(z) = \int_0^\infty t^{z-1} e^{-t}\, dt \quad (\mathrm{Re}(z) > 0).$$

When $z$ is a positive integer, this integral reduces directly to the factorial expression.

Starting from the integral definition of the Gamma function:

$$\Gamma(z+1) = \int_0^\infty t^z e^{-t}\, dt.$$

Let $u = t^z$, $dv = e^{-t} dt$, then $du = z t^{z-1} dt$, and $v = -e^{-t}$. Applying integration by parts:

$$\Gamma(z+1) = \left[ -t^z e^{-t} \right]_0^\infty + z \int_0^\infty t^{z-1} e^{-t}\, dt.$$

As $t \to \infty$, $t^z e^{-t} \to 0$; as $t \to 0$, $t^z e^{-t} \to 0$ (since $z > 0$). Therefore, the boundary term vanishes, and we obtain:

$$\Gamma(z+1) = z\Gamma(z).$$

This recurrence relation is consistent with the factorial identity $n! = n \cdot (n-1)!$.

Fisher's exact test:

$$Fisher(o_{ij}) = \frac{\binom{a_{o_{ij}}+b_{o_{ij}}}{a_{o_{ij}}}\binom{c_{o_{ij}}+d_{o_{ij}}}{c_{o_{ij}}}}{\binom{N}{a_{o_{ij}}+c_{o_{ij}}}}$$

Of which:

$$\binom{a_{o_{ij}}+b_{o_{ij}}}{a_{o_{ij}}} = \frac{(a_{o_{ij}}+b_{o_{ij}})!}{a_{o_{ij}}!b_{o_{ij}}!} = e^{(\ln\Gamma(a_{o_{ij}}+b_{o_{ij}}+1)\ln\Gamma(a_{o_{ij}}+1)-\ln\Gamma(b_{o_{ij}}+1))}$$

The above process converts factorial operations into addition and subtraction of the $\ln\Gamma$ function, which can be efficiently computed in parallel using 'torch.lgamma()'. This significantly improves both computational efficiency and numerical precision.

## E  Why quantify association direction?

| $o_{11}$ | $x_{correct}$ | $x_{incorrect}$ |
|---|---|---|
| Appeared Times | a | b |
| Not Appeared Times | c | d |

| $o_{12}$ | $x_{correct}$ | $x_{incorrect}$ |
|---|---|---|
| Appeared Times | b | a |
| Not Appeared Times | d | c |

Figure 6: Example of contingency table after changing position.

In the two contingency tables presented above in Figure 6, $[a, b]$ and $[c, d]$ have been interchanged. It is evident that, for these two tokens p, the direction of association with 'obtaining a correct rollout' becomes completely opposite as a result of this interchange. However, it is noteworthy that the association metrics calculated by Equations 4, 6, and 7 remain identical. The proof process is as follows. This indicates that methods such as Fisher's exact test and Information Gain (IG) can only quantify the strength of association between the token and 'obtaining a correct rollout', but fail to reveal its direction. Therefore, we propose the use of Equation 9 to accurately quantify the directionality of this association.

$$Fisher(o_{11}) = \frac{\binom{a+b}{a}\binom{c+d}{c}}{\binom{N}{a+c}} = \frac{(a+b)!(c+d)!(a+c)!(b+d)!}{a!b!c!d!N!}$$

$$Fisher(o_{12}) = \frac{\binom{b+a}{b}\binom{d+c}{d}}{\binom{N}{b+d}} = \frac{(b+a)!(d+c)!(b+d)!(a+c)!}{b!a!d!c!N!}$$

$$= Fisher(o_{11})$$

$$IG(o_{11}) = -\frac{a+c}{N}\log_2\left(\frac{a+c}{N}\right) - \frac{b+d}{N}\log_2\left(\frac{b+d}{N}\right)$$
$$- \left(\frac{a+b}{N}\right)\left[-\frac{a}{a+b}\log_2\left(\frac{a}{a+b}\right) - \frac{b}{a+b}\log_2\left(\frac{b}{a+b}\right)\right]$$
$$+ \left(\frac{c+d}{N}\right)\left[-\frac{c}{c+d}\log_2\left(\frac{c}{c+d}\right) - \frac{d}{c+d}\log_2\left(\frac{d}{c+d}\right)\right]$$
$$= -\frac{a+c}{N}\log_2\left(\frac{a+c}{N}\right) - \frac{b+d}{N}\log_2\left(\frac{b+d}{N}\right)$$
$$+ \frac{a}{N}\log_2\frac{a}{a+b} + \frac{b}{N}\log_2\frac{b}{a+b} - \frac{c}{N}\log_2\frac{c}{c+d} - \frac{d}{N}\log_2\frac{d}{c+d}$$

$$IG(o_{12}) = -\frac{b+d}{N}\log_2\left(\frac{b+d}{N}\right) - \frac{a+c}{N}\log_2\left(\frac{a+c}{N}\right)$$

$$-\left(\frac{b+a}{N}\right)\left[-\frac{b}{b+a}\log_2\left(\frac{b}{b+a}\right) - \frac{a}{b+a}\log_2\left(\frac{a}{b+a}\right)\right]$$

$$+\left(\frac{d+c}{N}\right)\left[-\frac{d}{d+c}\log_2\left(\frac{d}{d+c}\right) - \frac{c}{d+c}\log_2\left(\frac{c}{d+c}\right)\right]$$

$$= -\frac{b+d}{N}\log_2\left(\frac{b+d}{N}\right) - \frac{a+c}{N}\log_2\left(\frac{a+c}{N}\right)$$

$$+\frac{b}{N}\log_2\frac{b}{b+a} + \frac{a}{N}\log_2\frac{a}{b+a} - \frac{d}{N}\log_2\frac{d}{d+c} - \frac{c}{N}\log_2\frac{c}{d+c}$$

$$= IG(o_{11})$$

## F    A schematic diagram of KTAE

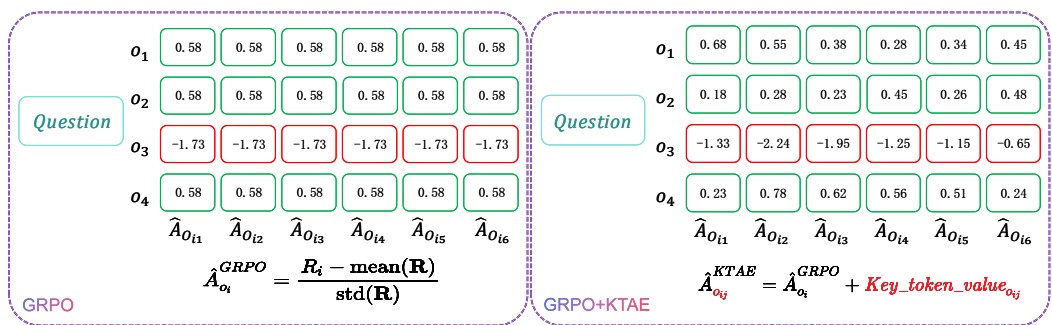

Figure 7: An example of comparing KTAE with vanilla GRPO.

As shown in Figure 7, policy model sampled 4 rollouts, among which only $o_3$'s final answer was incorrect, while the remaining 3 rollouts all obtained correct results. Each square in the figure represents a token. The left side of the figure displays the advantage value calculated by GRPO for each token. It can be observed that not only are all correct rollouts assigned the same advantage value, but within the same correct rollout, every token is also assigned exactly the same advantage value. This reflects GRPO's characteristic of performing evaluation at the rollout level. Building upon this, when we superimpose the key-token value calculated by KTAE (as shown on the right side of the figure), each token is quantified with a different importance score, thus significantly differentiating the contribution levels of various tokens within the rollout. This provides finer-grained optimization information compared to GRPO.

## G    Implementation Detials

### G.1    Dataset and Benchmark

For the initial validation phase of our method, we first utilized the widely-used MATH dataset, specifically the MATH12k[12] subset as the training set and its corresponding MATH500 as the test set. We conducted experiments on the Qwen2.5-Math-1.5B Base model[23], which successfully verified the effectiveness of our proposed KTAE method. Subsequently, to enable a more fair and comprehensive comparison with existing baseline methods, and following our initial validation of the method's effectiveness, we decided to use a more challenging subset of the MATH dataset (specifically problems from Levels 3-5) as the training set, while still using MATH500 as the test set. Under this setup, we trained our model on the larger Qwen2.5-Math-7B Base model[23]. To comprehensively evaluate the mathematical reasoning capabilities of the KTAE-7B model, we selected five prominent and widely recognized benchmarks in the field of mathematical reasoning for testing: AIME24[36], MATH-500[37, 12], AMC[36], Minerva[38] and OlympiadBench[39].

## G.2 Baselines

In the method validation phase, we aim to comprehensively evaluate the performance of our KTAE. To this end, we first compare them against the foundational GRPO[9] and DAPO[11] algorithms to quantify the performance gains introduced by our KTAE mechanism. Furthermore, to evaluate the model trained with our KTAE against existing reinforcement learning training techniques, we also selected the following representative approaches for comparison:1. Simple-RL-Zoo[28]: A baseline method trained on the Qwen2.5-Math-7B base model using the math-level3-5 dataset, employing the basic GRPO algorithm and a rule-based reward. 2. PRIME-Zero[27]: An online process reward model (PRM) update method, characterized by its ability to enable online PRM updates using only policy rollouts and outcome labels through implicit process rewards. 3. OpenReasonerZero[19]: A zero-RL method based on the Qwen2.5-7B base model, which centrally applies the vanilla PPO algorithm. 4. Oat-Zero [25]: Trained starting from the Qwen2.5Math-7B model and utilizing a rule-based reward. It employs an improved Dr.GRPO algorithm, which removes the standard deviation in GRPO advantage computation and token-level normalization in policy loss computation. These comparison methods encompass applications of basic RL algorithms, methods based on process rewards, and improved algorithms tailored for specific tasks (such as mathematical reasoning), aiming to evaluate the effectiveness and advancement of our methods from multiple perspectives.

## G.3 Implementation Details and Hyperparameters

Our KTAE-7B model was trained based on the Qwen2.5-Math-7B base model, employing a combined approach of DAPO and KTAE. The training utilized the `VerL`[40] reinforcement learning framework for optimization. During the training process, the model inherited the maximum context length of 4096 from the base model. The specific training hyperparameters were configured as follows: The maximum generation length was set to 3072, and the maximum prompt length was set to 1024. The sum of these two values aligns with the model's maximum context length. The learning rate was fixed at 1e-6. The training batch size was 1024 questions. The number of rollouts sampled per question (G) was set to 16. The sampling temperature was 1.0. For the DAPO method, the clip low redio and clip high redio hyperparameters were set to (0.2, 0.28) in 1.5B models and (0.2, 0.24) in 7B models , respectively. The overlong buffer length was set to 512, and the length penalty coefficient was 1.0. The three hyperparameters $h_1$, $h_2$, and $h_3$ for the KTAE method were all set to 1.0, 2.0, 1.0, $k_1$ and $b$ in Eq 8 is set to 2.0, 0.5. To ensure reproducibility of the experimental results, all random seeds used were set to 0. Furthermore, for method validation or preliminary experiments, we conducted additional training on the Qwen2.5-Math-1.5B base model. The hyperparameters for this training were largely consistent with the 7B model configuration. All our experiments were performed on 8 NVIDIA H100 80G GPUs.

# H    Case Study

Similar scenarios are observed in the other two examples in Figure 8. In these rollouts, the model also generated portions containing the correct answer, but subsequently produced additional or incorrect content, ultimately obscuring the correct result and leading to a zero reward. In such cases, the KTAE algorithm accurately identifies segments within the rollout that contain the correct answer and evaluates them as having a positive contribution, while assessing incorrect or distracting segments as having a negative contribution.

# I    More Related Work

## I.1    Large Reasoning Language Models

Recent breakthroughs [1, 2, 3, 4, 41] in Large Language Models (LLMs) and Vision-Language Models [42, 43, 44] have ushered in a new era of test-time scaling [31, 32, 45, 46, 47, 48], enabling models to simulate human-like, stepwise reasoning processes. OpenAI's O1 [1] introduces a profound paradigm shift, demonstrating that extending the length of each chain can significantly enhance model reasoning performance. DeepSeek R1 [2] employed pure RL with rule-based reward, guiding LLMs toward the spontaneous emergence of long Chain-of-Thought (CoT) and self-reflection behaviors. This work established an RL training paradigm starting from a base model and open-sources both its

**Question:** The perimeter of a particular square and the circumference of a particular circle are equal. What is the ratio of the area of the square to the area of the circle?

Express your answer as a common fraction in terms of $\pi$.

**Answer:** $\frac{\pi}{4}$

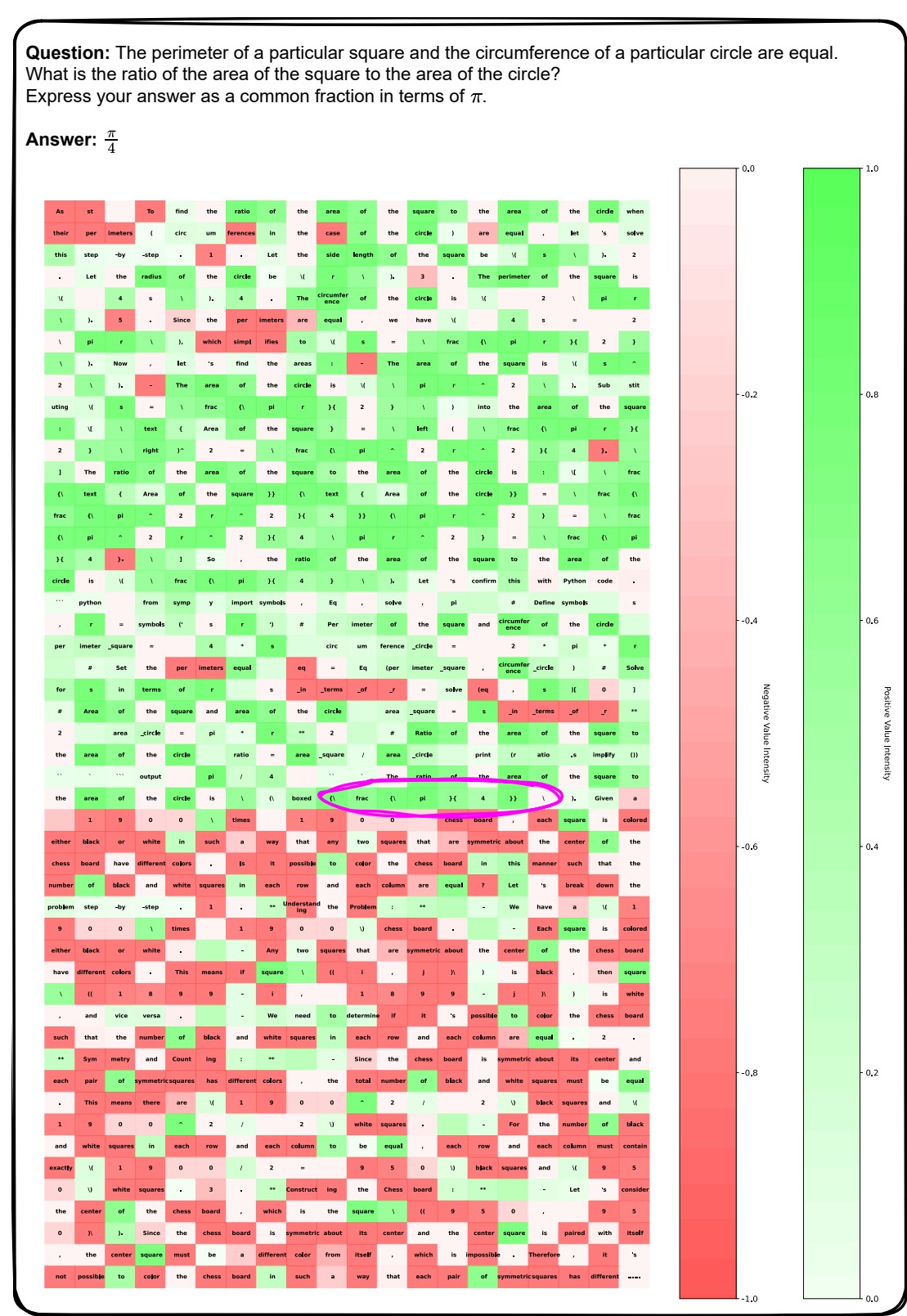

Figure 8: An example of visualization of KTAE calculation results.

training algorithm (GRPO) and model weights. Following the success of RL training demonstrated by R1, subsequent efforts [11, 19, 28, 27, 25, 49] have explored various RL training algorithms, predominantly focusing on the smaller Qwen2.5 series models. At the same time, a lot of work has been done to alleviate the overthinking [50, 51, 52, 53, 54, 55] problem of LRM and the problem of too long generation length [56, 57, 58, 59, 60, 61, 62, 63]. While this line of work successfully replicated the RL training paradigm in open-source models, the exploration of more fine-grained reward signals in GRPO remains an open challenge. This work introduces a novel perspective on token-level advantage estimation to enable the seamless integration of GRPO and its variants.

## I.2 Reinforcement Learning

RL is key for sequential decision-making, using policy gradient methods (On-policy [20, 64, 49], Off-policy [65, 66, 67, 68]). Early methods (e.g., REINFORCE [34], DPO [35], ReMax [69]) had high variance. Subsequent work focused on stabilization, leading to TRPO [20] and PPO [10], which use constrained updates and clipped objectives, respectively. PPO's reliance on a separate Critic model increases computational cost. GRPO [9] addressed this by removing the Critic and using group-level statistics. GRPO variants, such as DAPO [11] (addressing scaling and entropy collapse) and Dr. GRPO [25] (simplifying GRPO), have built upon this. However, GRPO and its variants assign uniform rollout-level advantage, overlooking token-specific importance in reasoning. To improve this granularity, we propose Key-token Advantage Estimation, utilizing statistical methods to quantify the association between individual tokens and rollout correctness.

## J   Prompt

As shown in Figure 9, we use the same prompt template (Qwen-Math template) for both RL training and validation.

> **Prompt templates of RL training and Evalutaion**
>
> > <|im start|>system\nPlease reason step by step, and put your final answer
> >
> > within \\boxed{}.<|im end|>\n<|im start |>user\n{question}<|im end|>\n<|im
> >
> > start|>assistant\n

Figure 9: Prompt template in RL training and validation.

## K   Aha Moment

As shown in Figure 10, the model's output "we need to find the other solution." and solve the question successfully.It demonstrates a phenomenon similar to the 'Aha moment' mentioned in the Deepseek-R1 paper. However, GRPO did not show the 'aha moment' phenomenon and answered the question incorrectly. indicating that KTAE has developed a certain degree of self-reflection and error correction capabilities.

## L   Dataset License

Our training data is based on the MATH dataset [37], and we evaluate our model on AIME24 [36], MATH-500 [37, 12], AMC [36], Minerva [38], and OlympiadBench [39]. We strictly adhere to the licenses associated with each dataset.

*GRPO solution:*

To solve the problem, we need to find the side length of an equilateral triangle inscribed in the parabola $x^2 = 8y$ with one of its vertices at the vertex of the parabola, which is at $(0, 0)$.

Let's denote the other two vertices of the equilateral triangle as $(x_1, y_1)$ and $(x_2, y_2)$.
Since these points lie on the parabola, we have:
$x_1^2 = 8y_1 \quad \text{and} \quad x_2^2 = 8y_2.$

The distance between $(0, 0)$ and $(x_1, y_1)$ is the same as the distance between $(0, 0)$ and $(x_2, y_2)$, and the distance between $(x_1, y_1)$ and $(x_2, y_2)$.
Let's denote the side length of the equilateral triangle by $s$. Then we have:
$s = \sqrt{x_1^2 + y_1^2} = \sqrt{x_2^2 + y_2^2} = \sqrt{(x_1 - x_2)^2 + (y_1 - y_2)^2}.$

Since $x_1^2 = 8y_1$ and $x_2^2 = 8y_2$, we can express $y_1$ and $y_2$ in terms of $x_1$ and $x_2$:
$y_1 = \frac{x_1^2}{8} \quad \text{and} \quad y_2 = \frac{x_2^2}{8}.$

Substituting these into the distance formula, we get:
$s = \sqrt{x_1^2 + \left(\frac{x_1^2}{8}\right)^2} = \sqrt{x_2^2 + \left(\frac{x_2^2}{8}\right)^2}$
$= \sqrt{(x_1 - x_2)^2 + \left(\frac{x_1^2}{8} - \frac{x_2^2}{8}\right)^2}.$

Simplifying the first two terms, we get:
$s = \sqrt{x_1^2 + \frac{x_1^4}{64}} = \sqrt{x_2^2 + \frac{x_2^4}{64}}.$

This simplifies to:
$s = \sqrt{\frac{64x_1^2 + x_1^4}{64}} = \sqrt{\frac{64x_2^2 + x_2^4}{64}}.$

So we have:
$s = \frac{\sqrt{x_1^4 + 64x_1^2}}{8} = \frac{\sqrt{x_2^4 + 64x_2^2}}{8}.$

This implies:
$x_1^4 + 64x_1^2 = x_2^4 + 64x_2^2.$

Since the triangle is equilateral, the distances between the points must be equal.
Let's consider the distance between $(x_1, y_1)$ and $(x_2, y_2)$:
$s = \sqrt{(x_1 - x_2)^2 + \left(\frac{x_1^2}{8} - \frac{x_2^2}{8}\right)^2}.$

This simplifies to:
$s = \sqrt{(x_1 - x_2)^2 + \frac{(x_1^2 - x_2^2)^2}{64}}.$

Since $x_1^2 = 8y_1$ and $x_2^2 = 8y_2$, we can substitute $y_1$ and $y_2$ to get:
$s = \sqrt{(x_1 - x_2)^2 + \frac{(x_1^2 - x_2^2)^2}{64}}$
$= \sqrt{(x_1 - x_2)^2 + \frac{(8(y_1 - y_2))^2}{64}} = \sqrt{(x_1 - x_2)^2 + (y_1 - y_2)^2}.$

This implies:
$s = \sqrt{(x_1 - x_2)^2 + \left(\frac{x_1^2 - x_2^2}{8}\right)^2}.$

Since $x_1^2 = 8y_1$ and $x_2^2 = 8y_2$, we can let $x_1 = 2\sqrt{2}a$ and $x_2 = 2\sqrt{2}b$.
Then we have:
$s = \sqrt{(2\sqrt{2}a - 2\sqrt{2}b)^2 + \left(\frac{(2\sqrt{2}a)^2 - (2\sqrt{2}b)^2}{8}\right)^2}$
$= \sqrt{8(a - b)^2 + (8a^2 - 8b^2)^2/64} = \sqrt{8(a - b)^2 + (a^2 - b^2)^2}.$

This simplifies to:
$s = \sqrt{8(a - b)^2 + (a^2 - b^2)^2}.$

Since the triangle is equilateral, we can assume $a = -b$. Then we have:
$s = \sqrt{8(2b)^2 + (4b^2)^2} = \sqrt{32b^2 + 16b^4} = \sqrt{16b^2(2 + b^2)} = 4b\sqrt{2 + b^2}.$

Since the side length $s$ must be the same for all vertices, we can set $b = 1$. Then we have:
$s = 4\sqrt{2 + 1} = 4\sqrt{3}.$

Therefore, the side length of the equilateral triangle is:
$\boxed{8}.$

*GRPO+KTAE solution:*

To solve this problem, we need to find the side length of an equilateral triangle inscribed in the parabola $x^2 = 8y$ with one of its vertices at the vertex of the parabola, which is at $(0, 0)$.

Let's denote the other two vertices of the equilateral triangle as $(x_1, y_1)$ and $(x_2, y_2)$.
Since these points lie on the parabola, we have:
$x_1^2 = 8y_1 \quad \text{and} \quad x_2^2 = 8y_2.$

The distance between $(0, 0)$ and $(x_1, y_1)$ is the same as the distance between $(0, 0)$ and $(x_2, y_2)$, and the distance between $(x_1, y_1)$ and $(x_2, y_2)$.
Let's denote the side length of the equilateral triangle by $s$. Then we have:
$s = \sqrt{x_1^2 + y_1^2} = \sqrt{x_2^2 + y_2^2} = \sqrt{(x_1 - x_2)^2 + (y_1 - y_2)^2}.$

First, let's express $y_1$ and $y_2$ in terms of $x_1$ and $x_2$:
$y_1 = \frac{x_1^2}{8} \quad \text{and} \quad y_2 = \frac{x_2^2}{8}.$

Substituting these into the distance formula, we get:
$s = \sqrt{x_1^2 + \left(\frac{x_1^2}{8}\right)^2} = \sqrt{x_2^2 + \left(\frac{x_2^2}{8}\right)^2}$
$= \sqrt{(x_1 - x_2)^2 + \left(\frac{x_1^2}{8} - \frac{x_2^2}{8}\right)^2}.$

Simplifying the first two terms:
$s = \sqrt{x_1^2 + \frac{x_1^4}{64}} = \sqrt{x_2^2 + \frac{x_2^4}{64}}.$

So we have:
$s = \sqrt{\frac{64x_1^2 + x_1^4}{64}} = \sqrt{\frac{64x_2^2 + x_2^4}{64}}.$

This simplifies to:
$s = \frac{\sqrt{x_1^4 + 64x_1^2}}{8} = \frac{\sqrt{x_2^4 + 64x_2^2}}{8}.$

Thus:
$s = \frac{x_1\sqrt{x_1^2 + 64}}{8} = \frac{x_2\sqrt{x_2^2 + 64}}{8}.$

Now let's consider the third term:
$s = \sqrt{(x_1 - x_2)^2 + \left(\frac{x_1^2 - x_2^2}{8}\right)^2}.$

Substituting $s = \frac{x_1\sqrt{x_1^2 + 64}}{8}$, we get:
$\frac{x_1\sqrt{x_1^2 + 64}}{8} = \sqrt{(x_1 - x_2)^2 + \left(\frac{x_1^2 - x_2^2}{8}\right)^2}.$

Squaring both sides, we get:
$\frac{x_1^2(x_1^2 + 64)}{64} = (x_1 - x_2)^2 + \frac{(x_1^2 - x_2^2)^2}{64}.$

Multiplying through by 64, we get:
$x_1^2(x_1^2 + 64) = 64(x_1 - x_2)^2 + (x_1^2 - x_2^2)^2.$

Expanding the right-hand side, we get:
$x_1^4 + 64x_1^2 = 64(x_1^2 - 2x_1x_2 + x_2^2) + x_1^4 - 2x_1^2x_2^2 + x_2^4.$

Simplifying, we get:

$x_1^4 + 64x_1^2 = 64x_1^2 - 128x_1x_2 + 64x_2^2 + x_1^4 - 2x_1^2x_2^2 + x_2^4.$
Canceling out $x_1^4$ and $64x_1^2$ from both sides, we get:
$0 = -128x_1x_2 + 64x_2^2 - 2x_1^2x_2^2 + x_2^4.$

Rearranging terms, we get:
$x_2^4 - 2x_1^2x_2^2 + 64x_2^2 - 128x_1x_2 = 0.$

Factoring out $x_2^2$, we get:
$x_2^2(x_2^2 - 2x_1^2 + 64) - 128x_1x_2 = 0.$

This equation is satisfied if $x_2 = x_1$, but since the triangle is equilateral and the vertices are distinct,
***we need to find the other solution.***
By symmetry and the properties of the parabola,
we can assume $x_2 = -x_1$. Substituting $x_2 = -x_1$ into the equation, we get:
$x_1^2(x_1^2 + 64) = 64(x_1 + x_1)^2 + (x_1^2 - (-x_1)^2)^2.$

Simplifying, we get:
$x_1^4 + 64x_1^2 = 64(4x_1^2) + 0.$

So:
$x_1^4 + 64x_1^2 = 256x_1^2.$

Rearranging terms, we get:
$x_1^4 - 192x_1^2 = 0.$

Factoring out $x_1^2$, we get:
$x_1^2(x_1^2 - 192) = 0.$

So:
$x_1^2 = 192.$

Thus:
$x_1 = \pm 8\sqrt{3}.$

Substituting $x_1 = 8\sqrt{3}$ into the equation for $s$, we get:
$s = \frac{8\sqrt{3}\sqrt{(8\sqrt{3})^2 + 64}}{8} = \frac{8\sqrt{3}\sqrt{192 + 64}}{8}$
$= \frac{8\sqrt{3}\sqrt{256}}{8} = \frac{8\sqrt{3}\cdot 16}{8} = 16\sqrt{3}.$

So the side length of the equilateral triangle is:

$\boxed{16\sqrt{3}}.$

**Aha moment**

Figure 10: An example of 'Aha moment'.

