# OpenReview forum: "KTAE: A Model-Free Algorithm to Key-Tokens Advantage Estimation in Mathematical Reasoning"
_NeurIPS.cc/2025/Conference — NeurIPS 2025 poster_

### Official Review · Reviewer_BQ9i · 2025-07-03

**Clarity:** 3
**Significance:** 3
**Originality:** 3
**Rating:** 4
**Confidence:** 4

**Summary:**

Key-Token Advantage Estimation (KTAE) tackles the coarse-granularity problem of existing RLHF variants (e.g., GRPO, DAPO) by computing fine-grained, token-level advantages instead of assigning a single rollout-level value to every token. The method statistically infers each token’s contribution to the final correctness signal and blends this with the rollout-level advantage, remaining model-free and easy to plug into standard policy-gradient pipelines. Empirically, GRPO+KTAE and DAPO+KTAE outperform their baselines on five mathematical-reasoning benchmarks, achieving higher accuracy with shorter answers and even surpassing R1-Distill-Qwen-1.5B when using the same backbone.

**Questions:**

Please consider the weaknesses outlined above; if subsequent responses offer convincing clarifications, I will be happy to revisit and potentially raise my score.

**Ethical Concerns:**

["NO or VERY MINOR ethics concerns only"]

**Final Justification:**

The authors provided reasonable clarifications for my main concerns, including the rationale for the statistic composition and the causes of benchmark regressions. While these do not fully resolve all limitations, I found the responses satisfactory, so I maintain my borderline accept recommendation.

**Limitations:**

yes

**Quality:**

3

**Strengths And Weaknesses:**

## Strengths
- **Fine-grained, token-level credit assignment:** Existing RLHF-style methods propagate a single rollout-level advantage to every token, so they cannot tell which parts of the answer actually “earned” the reward. KTAE attacks this long-standing credit-assignment bottleneck by statistically estimating each token’s marginal contribution to the final correctness signal, giving the optimizer far more informative gradients.
- **Model-free and compute-efficient:** Unlike token-reward approaches that bolt on an auxiliary critic or a learned reward network, KTAE is purely analytical; it re-weights the existing rollout advantage with a closed-form importance estimate. That means no extra parameters, no new forward passes, and no critic-training instability, making it easy to drop into any policy-gradient pipeline. This lightweight property will appeal to practitioners who already find RLHF training expensive.
- **Consistent empirical gains on math-reasoning suites:** Across five benchmark datasets for mathematical reasoning, GRPO+KTAE and DAPO+KTAE deliver higher exact-answer accuracy while also shortening the generated solutions.

## Weaknesses
- **Heuristic, non-learned composition of statistics:** KTAE sidesteps a learned critic by multiplying four hand-selected statistics—Fisher’s test score, information gain, token frequency, and the rollout-level advantage—to obtain a key-token value. While this model-free choice keeps computation cheap, the paper offers no theoretical rationale for using a product, nor does it explore alternative compositions or automatic ways to learn the combination. The ablation study shows each component matters in isolation, but I am left unsure whether such composition structure is robust when ported to new domains.
- **Surrounding contexts are ignored when scoring tokens:** A token’s value can depend on where it appears (e.g., a numeral “2” inside a denominator vs. as a final answer). Because KTAE treats each token independently, it may mis-attribute credit when the semantic payoff arises only from multi-token spans or long-range dependencies. That limitation becomes more glaring in text-rich domains (dialogue, code) where meaning is rarely confined to a single token.
- **Sparse diagnosis of performance regressions:** Although KTAE generally boosts accuracy, it drops on two of the five benchmarks (AIME24 for GRPO+KTAE and OlympiadBench for DAPO+KTAE). The paper merely remarks that this is “one fewer correct answer” in a small test set, offering no deeper probe into why the estimator hurt — for instance, whether token-level weighting mis-fires on very short proofs, whether the length-penalty interacts with Olympiad-style solutions, or whether sampling variance alone explains the loss. A concise error taxonomy (e.g., mis-tagged key tokens vs. over-shortened outputs) plus a couple of annotated examples would clarify whether the regression stems from KTAE itself or dataset.
- **Domain scope is narrow:** All experiments revolve around deterministic, auto-gradable math problems. It is unknown whether KTAE remains effective in open-ended generation where rewards are noisy, delayed, or reference-based (e.g., ChatGPT-style helpfulness RLHF). A broader evaluation—or at least a discussion of potential adaptations—would strengthen claims of generality.

---

> ### Author Rebuttal · Authors · 2025-07-31
>
> Thank you for your valuable feedback on our work. We've carefully considered the weaknesses you raised and offer the following clarifications:
>
> **W1:** We detail the theoretical basis for using **multiplicative combination** in Appendix E. Contingency tables inherently reflect the *likelihood* of an association but don't specify *which* particular metric is associated. Therefore, to determine importance, we must calculate the **direction of association** and multiply it by the **degree of association**. While we acknowledge that learning to automatically determine combination methods could offer benefits, it would introduce additional training overhead. We plan to explore other combination approaches in future work.
>
> Importantly, our goal is to develop a model-free, low-cost, and interpretable mechanism for estimating token-level advantage, which avoids the instability and overhead of training additional reward or value models. KTAE's calculation method is purely **statistical** and entirely **task-domain independent**. As long as a binary reward system is used, KTAE remains applicable when transferred to other domains.
>
>
> **W2:** The core of the KTAE algorithm rests on the assumption that "if a token frequently appears in all **correct rollouts** and rarely in all **incorrect rollouts**, then that token is highly important for achieving correct rollouts." We believe this assumption aligns with common intuition, and KTAE effectively models it using statistical methods.
>
> We fully acknowledge that KTAE currently doesn't account for **token order** or **inter-token dependencies**. However, KTAE is fundamentally an advancement built upon GRPO. The original GRPO assigns uniform advantage to every token in a rollout, completely overlooking individual token differences, let alone their order or dependencies. KTAE represents a significant step towards **token-level advantage** within the GRPO framework, making it challenging to address all complexities simultaneously. Quantifying token order and dependencies is inherently difficult; even highly trained models struggle to precisely evaluate their impact on correct mathematical reasoning steps. We agree that incorporating these aspects would undoubtedly lead to more accurate token-level advantage, and this is a promising direction worth studying for future research.
>
> **W3:** Our experiments were exclusively trained on the **MATH dataset**, meaning performance on other benchmarks relies entirely on **domain generalization capabilities**. Reinforcement learning training involves extensive sampling, introducing inherent randomness. Consequently, achieving consistent improvements across all benchmarks is challenging. We consider **AIME24** and **OlympiadBench** to be among the most difficult benchmarks, also requiring the longest CoT (Chain of Thought) lengths. DAPO (Decision-aware Policy Optimization) intrinsically includes a length penalty. This led to DAPO+KTAE's length decreasing too rapidly on OlympiadBench, causing the model to prioritize brevity over some accuracy. For GRPO+KTAE on AIME24, the performance decrease was marginal, corresponding to just one fewer correct answer. We attribute such minor fluctuations to the **stochastic nature of reinforcement learning training**.
>
> We appreciate the suggestion for deeper analysis. We are preparing a brief taxonomy of error modes (e.g., missed key tokens, excessive length reduction) and will include annotated examples to clarify where KTAE succeeds or fails in the final version.
>
>
> **W4:** From a principled standpoint, KTAE is **task-domain agnostic**. It's entirely probability-based, making it applicable to any domain that uses **binary rewards** for **GRPO** reinforcement learning, such as code generation or question answering (QA). In fact, for any task in any domain where a reasonable binary reward objective can be designed, KTAE can be used to identify critical tokens. Even with continuous numerical rewards, KTAE can be applied by binarizing them through thresholding. We will discuss these potential adaptations more explicitly in the final version, and we consider generalizing KTAE beyond math reasoning a central goal of our future work.
>
>
> We hope these clarifications adequately address your concerns. We would be sincerely grateful if, upon reviewing our response, you would consider reconsidering your score and potentially increasing it.

---

> > ### Comment · Reviewer_BQ9i · 2025-08-04
> >
> > Thank you for the clarifications. I will maintain my original rating.

---

### Official Review · Reviewer_GKsR · 2025-07-03

**Clarity:** 3
**Significance:** 2
**Originality:** 2
**Rating:** 4
**Confidence:** 4

**Summary:**

This work studies the problem of doing reinforcement learning finetuning for large language models. It has been observed that many existing methods (e.g., GRPO and DAPO) typically assigns a uniform advantage function for the entire rollout, lacking more fine-grained granularity. This work proposes the KATE framework, which can be directly leveraged within GRPO and DAPO to provide fine-grained, token-level advantages without introducing additional models. In particular, KATE builds a contingency table and uses hypothesis testing for estimating the association between tokens and the final correctness, which is further complemented with the information-theory-based method using conditional entropy. Also, to determine the direction of the association, the idea of term frequency is also incorporated. With all these approaches combined, a final importance score is computed, which is used as the token-level advantage. The effectiveness of KATE is demonstrated via experiments through its combination with GRPO and DAPO on training models with different sizes.

**Questions:**

I would appreciate the authors' thoughts on the concerns that I raised in the weakness section, in particular, regarding whether the current method simplifies the token-correctness association too much (i.e., ignoring some potentially important factors), how the final score is connected to the advantage, and how the proposed method can be positioned beyond GRPO and DAPO.

**Ethical Concerns:**

["NO or VERY MINOR ethics concerns only"]

**Final Justification:**

The authors have elaborated their perspectives on the connection between the heuristics in this work and the advantage function, which facilities my understanding and leads to my raised score. However, I personally maintain a suspicious opinion, as the proposed methods are still largely heuristics.

**Limitations:**

Yes

**Quality:**

2

**Strengths And Weaknesses:**

I have a very mixed feeling about the proposed method, which are listed in the following.


Strength:

- It is indeed an interesting and important problem of how to assign more fine-grained training signals for the tokens in the rollout during RL finetuning of LLM. I appreciate the authors' valuable efforts in this direction.

- Many different ideas are incorporated and nicely combined. The final score handles both the strength of the association and the direction of the association.

- There is no additional model training required, and the proposed method can be easily incorporated with GRPO, DAPO, and other similar methods.

Weakness:
-  From one perspective, as mentioned above, it is carefully crafted with many different ideas incorporated. On the other hand, it is slightly against my intuition that "directly counting" the token occurrence and the conditional entropy can be a strong indication of the association of the token with the response correctiveness. In particular, it seems that the method ignores the order of the tokens and the connection between tokens, which may be important.

- Also, the constructed method is largely heuristic without many theoretical insights. In particular, the score will be served as an advantage function while its connection with the advantage function is unclear to me, which seems to be more of an artifact created based on intuitions.

- Furthermore, the proposed method seems to be mostly suitable for algorithms like GRPO and DAPO, which performs multiple rollouts from one prompt. It does not really apply for PPO and others (which uses value networks to estimate advantages). This may limit the usefulness of the proposed method. In particular, from a higher-perspective, to obtain a more fine-grained advantage estimation, it essentially (based on my understanding) viewing the problem from the single-step one (bandits) to a multiple-step one (MDP), and algorithms like PPO are well-desighed for the later case (although the current LLM training are mostly treated in a single-step manner)

---

> ### Author Rebuttal · Authors · 2025-07-31
>
> Thank you for your valuable feedback on our work. We've addressed the weaknesses you raised below:
>
> **W1:**  The core idea behind the **KTAE algorithm** is based on the assumption that "if a token frequently appears in all correct rollouts and rarely appears in all incorrect rollouts, then that token is very important for obtaining correct rollouts." We believe this assumption aligns with common understanding, and KTAE models it using statistical methods. While KTAE indeed doesn't account for the **order of tokens** or **dependencies between tokens**, which are crucial factors, it's essentially an improvement upon **GRPO**. The original GRPO assigns the same advantage to every token in a rollout, completely ignoring the differences between tokens, let alone their order or dependencies. KTAE takes a step towards **token-level advantage** based on GRPO, so it's challenging to cover every aspect. Furthermore, quantifying the order of tokens and their dependencies is incredibly difficult. Even trained models struggle to accurately assess the impact of token order and dependencies in mathematical reasoning steps on achieving correct rollouts. Of course, considering these factors would undoubtedly lead to more accurate token-level advantages, and we believe this requires continuous effort and experimentation in future work.
>
> **W2:**  The score computed by KTAE reflects a token’s estimated contribution to producing correct rollouts. We acknowledge that KTAE adopts a heuristic approach, but we would emphasize that it is a pragmatic solution to the pressing limitation of GRPO. Within the GRPO framework, we adopt the view that tokens with greater importance should receive higher advantage values, since they warrant stronger learning signals. This formulation provides a practical way to incorporate token-level discriminability into model-free RL training, complementing GRPO’s rollout-level estimation.
>
> **W3:**  Currently, KTAE is fundamentally an improvement on GRPO, refining GRPO's rollout-level advantage into a **token-level advantage**. Therefore, its applicability is limited to reinforcement learning training based on GRPO. The **PPO algorithm** doesn't require multiple samplings; it can obtain token-level advantage by treating each token as a step through a value model. It would be exciting if a **model-free algorithm** could calculate token-level advantage in PPO, and we will strive to achieve such an algorithm. Furthermore, KTAE is compatible with PPO-like methods. PPO typically uses a value model to compute per-token advantages, but these estimates are often coarse or noisy in reasoning tasks where rewards are sparse and delayed. KTAE can serve as a complementary module: its statistically grounded signals could be used to augment or denoise the token-level advantage estimates from a PPO critic. We leave a more thorough exploration of such integration to future work, as mentioned in the conclusion.
>
> We sincerely hope that our responses have addressed your concerns, and we would be grateful if you would consider updating your score and potentially increasing it.

---

> > ### Comment · Reviewer_GKsR · 2025-08-06
> >
> > I would like to thank the authors for kindly answering my question.
> >
> > All my concerns now boil down to one question (essentially the second point of weakness from my original reviews) -- how is the constructed estimation related to the advantage function? This is important but not fully addressed in the response. In particular, in RL, the advantage function itself has clear definitions and physical meanings, while this work, claiming to perform advantage estimation, does not provide any discussions on the relationship between the estimated value obtained via the proposed heuristics and the advantage.

---

> > > ### Author Response · Authors · 2025-08-06
> > >
> > > Dear Reviewer GKsR,
> > >
> > > We thank you for your insightful comment. We agree that the connection between our proposed estimation and the formal advantage function deserves a more rigorous theoretical discussion. Our KTAE is not intended to replace the standard advantage function but to act as a fine-grained, token-level correction term that addresses the coarse granularity of GRPO's rollout-level advantage. We present a derivation below to formalize this relationship.
> > >
> > > **1. Positioning KTAE: A Fine-Grained Modulator for the GRPO Advantage Estimate**
> > >
> > > In reinforcement learning, the advantage function is defined as $A^{\pi}(s_t, a_t) = Q^{\pi}(s_t, a_t) - V^{\pi}(s_t)$. For text generation, where a reward $R(o)$ is given only at the end of a rollout $o=(o_{1},...,o_{t}​)$, the advantage of taking action ot​ in state st​ is the expected future reward of that action minus the average expected reward from that state. GRPO simplifies this by assigning the same advantage value $\hat{A}^{GRPO}_i = \frac{R(o_i) - mean}{std}$, to every token $o\_{i,t}$ in a given rollout $o\_i$. This is the source of the coarse granularity we aim to fix.
> > >
> > > Our method is fundamentally an improvement upon a rule-based GRPO algorithm. A key characteristic of this rule-based GRPO is its absence of both a reward model and a value model. Instead, it assigns a binary, rollout-level reward of 0 or 1 based on whether the final answer of each rollout is correct.
> > >
> > > Consequently, according to the GRPO formula for calculating advantage, the advantage in rule-based GRPO is also at the rollout-level. This means that every token within the same rollout shares an identical advantage value. This approach differs from PPO's GAE (Generalized Advantage Estimation) algorithm, where each token is treated as an individual step.Therefore, the physical meaning of the advantage function in rule-based GRPO is the advantage of a rollout that receives a reward of 1 relative to a rollout that receives a reward of 0. A higher advantage value indicates that the rollout is a better choice, and its probability of occurrence should be increased.
> > >
> > > **The core KTAE term $σ(...) - 0.5$** builds upon this by answering a more nuanced question: "**Given that this solution was good/bad, which specific tokens contributed more significantly to that outcome?**" By performing statistical analysis across multiple sampled rollouts, KTAE identifies key tokens that are highly correlated with successful outcomes and assigns them a higher "importance" score. The physical meaning of the estimated value calculated by KTAE is the strength of a token's correlation with achieving a reward of 1. A higher estimated value signifies that the presence of that specific token is more likely to lead to a reward of 1. Thus, this token is a better choice compared to others, its advantage should be higher, and its probability of being generated should be increased.
> > >
> > > Therefore, the relationship between KTAE and the advantage function can be understood as follows: KTAE provides a **proxy advantage signal** that serves to **decompose** the macro, rollout-level advantage into micro, token-level contributions.
> > >
> > > **2. Justifying KTAE's Components: Connecting Heuristics to RL Principles**
> > >
> > > While we agree that KTAE is constructed from components that may seem heuristic, these intuitions are grounded in RL principles.
> > >
> > > *   **Fisher's Exact Test (F) / Information Gain (IG):** These metrics quantify the **statistical correlation** between the occurrence of a token and a successful outcome. In RL, a high-advantage (s, a) pair implies that taking action $a$ leads to a much higher expected return than the average policy. Analogously, a token that is strongly correlated with success is, in a statistical sense, an "action" that significantly increases the probability of a high reward. We use F and IG to **quantify this potential to increase the expected return**, serving as a proxy for its intrinsic value.
> > >
> > > *   **Directional Score ($D(o\_ij)$):** The advantage function is fundamentally a **relative** concept (how much better than average). Our directional score directly estimates a token's **discriminative power** by comparing its standardized frequency in correct versus incorrect rollouts. A token that appears frequently **only** in correct solutions is clearly more "advantageous" in a relative sense than a common token (e.g., "the," "is") that appears everywhere. This aligns with the "relative" nature of the advantage function.
> > >
> > > In summary, the combined score $(h_1×F+h_2×IG)×D$ is not an arbitrary artifact. The F and IG terms represent the **magnitude** of a token's contribution, while the D term represents its **direction** and **specificity/importance**. Their product jointly constructs a metric that reflects a token's **marginal contribution** to the final outcome, which is precisely the core information we seek from an advantage function.

---

> > > > ### Author Response · Authors · 2025-08-08
> > > >
> > > > Dear Reviewer GKsR,
> > > >
> > > > We hope this message finds you well.
> > > >
> > > > Thank you so much for your thorough review and valuable feedback on our paper. We have carefully considered all your comments and have provided more detailed and clear explanations in our revised rebuttal.
> > > >
> > > > With the discussion period nearing its end, we would like to politely inquire if our response has sufficiently addressed your concerns. If you have any further thoughts or require additional clarification, we would be more than happy to provide it.
> > > >
> > > > Thank you again for your time and dedicated engagement.
> > > >
> > > > Best regards,
> > > > The Authors

---

> > > > > ### Comment · Reviewer_GKsR · 2025-08-08
> > > > >
> > > > > Thank you for the follow-up response. I will update my score.

---

> > > > > > ### Author Response · Authors · 2025-08-08
> > > > > >
> > > > > > Dear Reviewer GKsR,
> > > > > >
> > > > > > Thank you so much for your positive feedback and for acknowledging our clarifications. We are very pleased to hear that our explanations have successfully addressed your concerns. Thank you for your valuable feedback and for taking the time to consider our rebuttal. Your detailed and insightful comments were instrumental in improving the quality of our work, and we are grateful for your constructive engagement.
> > > > > >
> > > > > > We truly appreciate your support.
> > > > > >
> > > > > > Sincerely,
> > > > > > The Authors

---

> ### Author Response · Authors · 2025-08-05
>
> Dear Reviewer GKsR,
>
> We hope this message finds you well.
>
> We are deeply grateful for your thorough review and acknowledgment of our work. We have provided detailed clarifications in response to your valuable feedback.
>
> As the discussion period draws to a close, we would like to hear your thoughts on our response, including whether it adequately addresses your concerns. If you have any updated thoughts, we would be grateful to hear them.
>
> Thank you again for your time and thoughtful engagement.
>
> Best regards,
>
> Authors

---

### Official Review · Reviewer_8J8v · 2025-07-03

**Clarity:** 2
**Significance:** 3
**Originality:** 3
**Rating:** 4
**Confidence:** 4

**Summary:**

This paper proposes KtAE, a model-free algorithm to estimate token-level advantage of model rollouts to provide finer-grained supervision signals in LLM reinforcement learning.

Specifically, KtAE constructs contingency tables based on the correctness of sampled rollouts and uses statistical methods (Fisher's exact test and Information Gain) to quantify each token's association with correct outcomes.

The proposed method can be empirically deployed on existing algorithms GRPO and DAPO to further improve model performance. Interestingly, the proposed method can not only improve the accuracy, but also speed up the inference by generating shorter responses.

**Questions:**

* How could KtAE be compared and combined with fine-grained reinforcement learning methods? For example, what if we have multiple reward models for different features and different granularities?
* The relationship between BM25-inspired frequency calculation and token importance is unclear. Could you elaborate further on this?
* What if some tokens appear rarely across rollouts? How often would such corner cases occur and impact the final performance?
* Could you provide timing comparisons between KtAE and corresponding baseline to discuss its scalability to larger models and batch sizes?

**Ethical Concerns:**

["NO or VERY MINOR ethics concerns only"]

**Final Justification:**

The authors have well addressed my concerns. I would like to keep my positive rating.

**Limitations:**

yes

**Quality:**

3

**Strengths And Weaknesses:**

### Strengths
* The idea of using statistical association measures to identify key tokens is innovative and economic. The empirical performance is also surprisingly appealing. The case study of "aha moment" as a result of KtAE provides the insight that such token-level advantage helps to elicit the self-reflection behaviors of LLMs.
* Given the improvement in both performance and efficiency (reduced response length), the proposed method demonstrates strong empirical results with reasonable cost, showing potential to be applied at scale.
* The detailed mathematical formulations and implementation details show technical maturity.

### Weaknesses
* While the empirical results are strong, KtAE is mainly a combination of statistical measures. Given the sparsity of token-level supervision signals (especially when rollout budget is limited), it is hard to quantify the quality and stability of the token-level advantages. The authors could either (i) provide detailed convergence analysis or theoretical guarantees about the quality of the token-level advantages; or (ii) conduct further ablation study to probe the sensitivity of KtAE to the statistical hyperparameters such as the number of rollouts and length of responses.
* The token-level advantages are mainly determined by the binary rewards and statistical estimation. This makes KtAE a bit limited to mathematical reasoning (which has determined final answers).
* KtAE introduces some additional hyperparameters (h1, h2, h3, k1, b) which may need to be adjusted on different datasets. It would be better to include some sensitivity analysis to help understand the generalizability and stability of the method.
* While the token-level advantage calculation does not require additional model calls, it would still be helpful to discuss the computational overhead from this process to understand the complexity of the method.

---

> ### Author Rebuttal · Authors · 2025-07-31
>
> Thank you for your valuable feedback on our work. We've addressed the weaknesses you raised below:
>
> **W1:**  The core idea behind the **KTAE algorithm** is based on the assumption that "if a token frequently appears in all correct rollouts and rarely appears in all incorrect rollouts, then that token is very important for obtaining correct rollouts." We believe this assumption aligns with common understanding, and KTAE models it using statistical methods.
>
> Currently, mainstream training methods typically limit the number of sampled rollouts per problem to 8 or 16, and our experiments follow this setup. In practice, a smaller 'n' value often leads to situations where all sampled rollouts for a given problem yield identical rewards, meaning no single rollout offers a significant advantage over others. In such cases, KTAE, like GRPO, calculates an advantage value of zero for all tokens. In a sense, the evaluation quality of KTAE is positively correlated with that of GRPO. Theoretically, a larger number of rollouts or a more uniform distribution of binary rewards among the 'n' rollouts, which is analogous to a larger statistical sample, would improve KTAE's evaluation quality. Your point is certainly worth investigating, but due to time and computational constraints, we may not be able to conduct a more fine-grained experimental analysis of KTAE in the short term. In future work, we will focus on analyzing KTAE's sensitivity to statistical hyperparameters.
>
> **W2:**  We believe that **KTAE** is fundamentally task-agnostic. It's entirely probability-based and can be applied to any domain that uses **binary rewards** for **GRPO reinforcement learning (RL) training**, such as code generation or question answering. The reason we've focused on mathematical reasoning tasks is simply that rule-based binary reward methods are prevalent in this area, and related work primarily conducts experiments here. This choice allows for easier comparison. Essentially, KTAE can be used to calculate key tokens for any task in any domain, provided a suitable binary reward optimization objective can be designed.
>
>
> **W3:**  While KTAE does introduce several hyperparameters, we didn't deliberately tune them in our practical use. We simply chose relatively average weight values for each component as hyperparameters, without specifically emphasizing any particular component. Since the principles behind each component are task-agnostic, we believe that these average weights theoretically don't need to be adjusted for different tasks. As you pointed out, KTAE has many hyperparameters, making a detailed sensitivity analysis for each parameter difficult in the short term. We performed a coarse-grained ablation study on each component in our paper, demonstrating their effectiveness. In the future, we will focus on analyzing KTAE's sensitivity to statistical hyperparameters.
>
> **W4&Q4:**  Regarding training efficiency, we conducted supplementary experiments. The average time per step using 8 H100 GPUs for training on the 7B and 1.5B models is as follows:
>
> |                            | Qwen2.5-7B-MATH/s | Qwen2.5-1.5B-MATH/s |
> |----------------------------|-----------------|-------------------|
> | GRPO                       | 559.36          | 277.69            |
> | GRPO+KTAE                  | 641.98          | 371.79            |
> | DAPO                       | 1006.30         | 363.36            |
> | DAPO+KTAE                  | 1159.00         | 642.74            |
>
> As you can see, **KTAE** is a model-free algorithm, and its computation time is almost independent of the model scale, depending only on the number of tokens generated. For the 7B model, due to its larger size, each training step takes longer, so the computational efficiency loss introduced by KTAE is not as noticeable. Conversely, for the 1.5B model, since the training steps are shorter, the training efficiency loss from KTAE is more pronounced. Nevertheless, KTAE's computational overhead is still acceptable compared to methods that utilize the model itself. In reality, the computation time for KTAE largely depends on code optimization. If only CPU serial computation were used, it would take several hours. The current code only performs small tensor parallel computations at the level of N rollouts in a single sampling, leading to a very low degree of data parallelism. We haven't yet thoroughly optimized the computational efficiency of the KTAE algorithm from an engineering perspective; GPU utilization when computing KTAE is less than 1%. We can definitely concatenate these into larger tensors, which would further leverage GPU capabilities and significantly reduce computation time. We will attempt further engineering optimization in the future.
>
> **Q1:**  If there were multiple reward models for different features and granularities, each rollout would receive multi-dimensional and diverse rewards. Theoretically, KTAE currently only applies to binary rewards. However, if the multiple rewards from different reward models can be combined and compressed into a **binary reward** based on task characteristics, then KTAE would still be applicable.
>
> **Q2:**  We believe that a token appearing in multiple correct rollouts doesn't fully represent its relevance to those correct rollouts. Its importance is truly indicated only when it frequently appears in correct rollouts and rarely appears in incorrect ones. Statistical values calculated solely from contingency tables only consider "whether it appeared," which can magnify the influence of low-frequency words. Therefore, we introduced **term frequency** to balance "whether it appeared" and "whether it appeared frequently."
>
> **Q3:**  If certain tokens appear with extremely low frequency across multiple rollouts, KTAE would still consider the impact of their presence or absence on obtaining a correct rollout. Even if they appear very rarely, if they appear in every correct rollout and in no incorrect rollouts, they would still receive a high advantage value. This applies to words in "aha moments," for example. While such situations are uncommon, KTAE accounts for their existence.
>
> We would be very grateful if our responses address your concerns and you would consider increasing our score!

---

> > ### Author Response · Authors · 2025-08-05
> >
> > Dear Reviewer 8J8v,
> >
> > We hope this message finds you well.
> >
> > We are deeply grateful for your thorough review and acknowledgment of our work. We have provided detailed clarifications in response to your valuable feedback.
> >
> > As the discussion period draws to a close, we would like to hear your thoughts on our response, including whether it adequately addresses your concerns. If you have any updated thoughts, we would be grateful to hear them.
> >
> > Thank you again for your time and thoughtful engagement.
> >
> > Best regards,
> >
> > Authors

---

> > ### Comment · Reviewer_8J8v · 2025-08-09
> >
> > Thanks for your detailed responses to address my concerns. My rating remains unchanged.

---

> > > ### Author Response · Authors · 2025-08-09
> > >
> > > Dear Reviewer 8J8v,
> > >
> > > Thank you for your valuable feedback and for taking the time to consider our rebuttal. Your detailed and insightful comments were instrumental in improving the quality of our work, and we are grateful for your constructive engagement.
> > >
> > > We truly appreciate your support.
> > >
> > > Sincerely,
> > >
> > > The Authors

---

### Official Review · Reviewer_3Vhm · 2025-07-04

**Clarity:** 3
**Significance:** 3
**Originality:** 3
**Rating:** 5
**Confidence:** 4

**Summary:**

This paper proposes KTAE: A Model-Free Algorithm to Key-Tokens Advantage Estimation in Mathematical Reasoning | OpenReview, a novel algorithm that estimates fine-grained, token-level advantages without introducing additional models. KTAE leverages the correctness of sampled rollouts and applies statistical analysis to quantify the importance of individual tokens within a sequence to the final outcome. Empirical results show that model trained with GRPO+KTAE and DAPO+KTAE outperform baseline methods across five mathematical reasoning benchmarks.

**Questions:**

- Can this approach generalize to tasks beyond mathematical reasoning?

**Ethical Concerns:**

["NO or VERY MINOR ethics concerns only"]

**Final Justification:**

The authors have well addressed my concerns. I would like to keep my positive rating.

**Limitations:**

YES

**Quality:**

3

**Strengths And Weaknesses:**

Pros:
- This paper is well-written and easy-to-understand. The idea is intuitive, reasonable, and effective.
- The importance of individual tokens within a sequence to the final outcome is a very important problem in reinforcement learning.
- Experiments are conducted based on GRPO and DAPO, two recent popular RL approaches for LLMs, proving the effectiveness of KTAE.
- Solid motivations and illustrations in Section 3.
- Clear visualization examples to show the important tokens found by KTAE.

Cons:
- Experiments are conducted on 1.54B/7B models. It is not clear whether this method also have the scaling law.
- It would be better if the paper can also analyze the training efficiency (comparing with baselines).

---

> ### Author Rebuttal · Authors · 2025-07-31
>
> Thank you very much for your valuable feedback on our work. We'd like to address the weaknesses you pointed out as follows:
>
> **W1:**  Currently, mathematical reasoning tasks commonly employ foundation models with a certain level of mathematical capability for reinforcement learning (RL) training. As a result, related work generally uses Qwen2.5 Math series models as base models, which are only available in 1.5B, 7B, and 72B sizes. Since RL training requires frequent sampling, it demands significant computational resources. Due to our current computational limitations, it's challenging for us to conduct experiments on larger models within a limited timeframe. In future work, we will continue to explore the effectiveness of KTAE on larger-scale models and models with different capabilities.
>
> **W2:** Regarding the training efficiency issue, we conducted supplementary experiments. The average time per step using 8 H100 GPUs for training on the 7B and 1.5B models is as follows:
>
> |                            | Qwen2.5-7B-MATH/s | Qwen2.5-1.5B-MATH/s |
> |----------------------------|-----------------|-------------------|
> | GRPO                       | 559.36          | 277.69            |
> | GRPO+KTAE                  | 641.98          | 371.79            |
> | DAPO                       | 1006.30         | 363.36            |
> | DAPO+KTAE                  | 1159.00         | 642.74            |
>
> As you can see, KTAE is a model-free algorithm, and its computational time is almost independent of the model scale, depending only on the number of tokens generated. For the 7B model, due to its larger size, each training step takes longer, so the computational efficiency loss introduced by KTAE is not as noticeable. Conversely, for the 1.5B model, since the training steps are shorter, the training efficiency loss from KTAE is more pronounced. Nevertheless, KTAE's computational overhead is still acceptable compared to methods that utilize the model itself. In reality, the computation time for KTAE largely depends on code optimization. If only CPU serial computation were used, it would take several hours. The current code only performs small tensor parallel computations at the level of N rollouts in a single sampling, leading to a very low degree of data parallelism. We haven't yet thoroughly optimized the computational efficiency of the KTAE algorithm from an engineering perspective; GPU utilization when computing KTAE is less than 1%. We can definitely concatenate these into larger tensors, which would further leverage GPU capabilities and significantly reduce computation time. We will attempt further engineering optimization in the future.
>
>
> **Q1:** Regarding your question, we believe that KTAE, in principle, is task-agnostic. It is entirely probability-based, and can be applied to any domain that uses binary rewards for GRPO reinforcement learning training, such as code generation or question answering. It just so happens that in the field of mathematical reasoning, rule-based binary reward methods are mainstream algorithms, and related work has been experimented with mathematical reasoning tasks. For ease of comparison, we also chose the same task for KTAE. In fact, KTAE can be used to calculate key tokens for any task in any domain, as long as a reasonable binary reward objective can be designed.
>
> We would be very grateful if our responses address your concerns and you would consider increasing our score!

---

### Decision · Program_Chairs · 2025-09-17

**Decision:**

Accept (poster)

**Comment:**

Existing reinforcement learning algorithms like GRPO and DAPO compute rollout-level advantages that assign the same value to all tokens in a sequence, overlooking token-specific contributions. To overcome this, the paper introduces Key-token Advantage Estimation (KTAE), a method that provides fine-grained, token-level advantage estimates without requiring additional models. KTAE works by analyzing the correctness of sampled rollouts and statistically quantifying each token’s contribution to the final outcome. This paper shows that algorithms combining GRPO and DAPO with KTAE outperform baseline methods across multiple benchmark tests. The idea is novel and the contribution is solid. Authors well address reviewers' concern during the rebuttal phase. We therefore recommend acceptance.